# Testing a novel sensor design to jointly measure cosmic-ray neutrons, muons and gamma rays for non-invasive soil moisture estimation

Stefano Gianessi[1*], Matteo Polo[2], Luca Stevanato[2], Marcello Lunardon[2,3], Till Francke[4], Sascha E. Oswald[4], Hami Said Ahmed[5], Arsenio Toloza[5], Georg Weltin[5], Gerd Dercon[5], Emil Fulajtar[6], Lee Heng[6], Gabriele Baroni[1]

[1] Department of Agricultural and Food Science, University of Bologna, Bologna, 40127, Italy
[2] FINAPP s.r.l., Montegrotto Terme (Padova), 35036, Italy
[3] Department of Physics and Astronomy, University of Padova, Padova, 35100, Italy
[4] Institute of Environmental Science and Geography, University of Potsdam, Potsdam, Germany
[5] Soil and Water Management and Crop Nutrition Laboratory, Joint FAO/IAEA Centre of Nuclear Techniques in Food and Agriculture Vienna, Austria
[6] Soil and Water Management and Crop Nutrition Section Joint FAO/IAEA Centre of Nuclear Techniques in Food and Agriculture Vienna, Austria
* now at FINAPP s.r.l., Montegrotto Terme (Padova), 35036, Italy

*Correspondence to*: Gabriele Baroni (g.baroni@unibo.it)

**Abstract.** Cosmic-ray neutron sensing (CRNS) has emerged as a reliable method for soil moisture and snow estimation. However, the applicability of this method beyond research has been limited due to, among others, the use of relatively large and expensive sensors. This paper presents the tests conducted to a new scintillator-based sensor especially designed to jointly measure neutron counts, muons and total gamma-rays. The neutron signal is firstly compared against two conventional gas-tube-based CRNS sensors at two locations. The estimated soil moisture is further assessed at four agricultural sites based on gravimetric soil moisture collected within the sensor footprint. Muon fluxes are compared to the incoming neutron variability measured at a neutron monitoring station and total gammas counts are compared to the signal detected by a gamma-ray spectrometer. The results show that the neutron dynamic detected by the new scintillator-based CRNS sensor is well in agreement with the conventional CRNS sensors. The derived soil moisture also agreed well with the gravimetric soil moisture measurements. The muons and the total gamma-rays simultaneously detected by the sensor show promising features to account for the incoming variability and for discriminating irrigation and precipitation events, respectively. Further experiments and analyses should be conducted, however, to better understand the accuracy and the added value of these additional data for soil moisture estimation. Overall, the new scintillator design shows to be a valid and compact alternative to conventional CRNS sensors for non-invasive soil moisture monitoring and to open the path to a wide range of applications.

## 1 Introduction

Soil moisture plays a key role in the hydrological cycle controlling water and energy fluxes at the land surface (Seneviratne et al., 2010; Vereecken et al., 2008). For this reason, an accurate monitoring of this variable is crucial in many applications, ranging from agricultural water management (Lichtenberg et al., 2015), runoff generation and floods (Bronstert et al., 2011; Saadi et al., 2020), and landslide prediction (Abraham et al., 2021; Zhuo et al., 2019). The main challenges in monitoring this variable are related to its strong spatial and temporal variability

driven by the different hydrological processes at the land surface (Haghighi et al., 2018) and further aggravated by human activities like irrigation and drainage (Domínguez-Niño et al., 2020).

Several instruments for monitoring soil moisture are nowadays available ranging from invasive point-scale soil moisture sensors to remote sensing methods with larger coverage (Babaeian et al., 2019; Corradini, 2014; Ochsner et al., 2013). More recently, attention has been paid to the development and assessment of the so-called proximal soil moisture sensors (Bogena et al., 2015). These non-invasive near-ground detectors have the advantages to estimate soil moisture over an intermediate scale (10 - 200 m radius) and at sub-daily resolutions providing a new perspective for hydrological observations (Ochsner et al., 2013).

Among these non-invasive techniques, cosmic-ray neutron sensing - CRNS (Zreda et al., 2008) - has shown good performance in several environmental conditions like natural ecosystems (Franz et al., 2012), meadow (Zhu et al., 2016), cropped fields (Rivera Villarreyes et al., 2011; Coopersmith et al., 2014), and forests (Heidbüchel et al., 2016; Jeong et al., 2021). This technique relies on the negative correlation between natural neutron fluxes in a specific energy range (0.5 eV – 100 keV) and hydrogen pools at and in the ground, providing the base for monitoring soil moisture (Zreda et al., 2012), snow (Schattan et al., 2017; Tian et al., 2016) and biomass (Baroni and Oswald, 2015; Jakobi et al., 2018).

Noteworthy, this negative correlation has been detected since long time but mostly considered nuisance in space weather monitoring (Hands et al., 2021; Hendrick and Edge, 1966) and rock dating (Gosse and Phillips, 2001). First studies showing the value of this signal for hydrological applications have been presented only some years later based on a neutron detector installed below the ground (Kodama et al., 1979). Its application, however, remained limited to some integrations into long-term observation networks for snow estimation (Morin et al., 2012). A strong contribution to the development and spread of this technique was provided only more recently when a better understanding of the interaction of these neutron fluxes and soil moisture was investigated (Zreda et al., 2008). In this context, the neutron detector had been installed above-ground and the signal well agreed with soil moisture over an area of several hectares and down to a depth of several decimetres (Franz et al., 2012; Köhli et al., 2015) providing a new prospective to monitor hydrological variables at the land surface (Desilets et al., 2010). Nowadays, this above-ground CRNS method is used by many research groups worldwide and it is integrated into some national monitoring systems for providing a better understanding of hydrological processes and supporting water management and assessments (Andreasen et al., 2017b; Bogena et al., 2022; Cooper et al., 2021; Hawdon et al., 2014; Upadhyaya et al., 2021; Zreda et al., 2012; Evans et al., 2016).

Initially, all the CRNS detectors were based on proportional gas tubes filled in with helium-3 or boron trifluoride (Schrön et al., 2018; Zreda et al., 2012). Alternative sensors are now emerging that could also open the path to new and wider applications (Cirillo et al., 2021; Flynn et al., 2021; Patrignani et al., 2021; Stevanato et al., 2019; Stowell et al., 2021; Weimar et al., 2020; van Amelrooij et al., 2022). In this context, the scintillator-based neutron detector design showed a good capability to measure neutrons with different energies (Cester et al., 2016). A first prototype specifically for soil moisture estimation was developed and tested showing good performance in comparison with independent soil moisture observations (Stevanato et al., 2019). This detector was further improved by, e.g., reducing environmental temperature effects on the recorded signal and reducing its energy consumption (Stevanato et al., 2020). First comparisons with independent data confirmed the good performances of these devices (Gianessi et al., 2021) with the additional advantage of measuring muons for on-site incoming neutron correction (Stevanato et al., 2022).

In this study, we present a comprehensive description and assessment of this new scintillator-based CRNS detector. The assessment is performed based on: (i) a comparison of the detected neutron counts with conventional gas-tube-based CRNS instruments at two experimental sites, (ii) a comparison of the derived soil moisture with independent gravimetric soil moisture measurements at four additional experimental sites, (iii) a comparison of detected muons with incoming neutrons measured at a neutron monitoring station, (iv) a comparison of total gamma counts with a conventional gamma ray spectrometer at one experimental site. The added value of muons and gamma particles simultaneously recorded by the sensor are also explored and discussed.

## 2 Materials and methods

### 2.1 The detector assembly

Scintillators have been identified as a promising alternative to proportional gas tubes for measuring neutrons in many applications (Peerani et al., 2012). The main advantages are the use of cheaper and safer materials than proportional gas tubes based on helium-3 or boron trifluoride, respectively. Moreover, the flexibility in manipulating the detecting material (e.g., thin layers) allows to optimize the sensitive area and to develop relatively efficient but compact sensors. The scintillators are made of plastic or organic materials that emit photons in the visible or near ultraviolet (UV) region when hit by radiation. The scintillator materials used for neutron detection, in particular, have the unique property in comparison to inorganic scintillator to release the light in different ways when hit by different particles. The identification of the type of particle or ray is achieved by means of Pulse Shape Analysis (PSA), exploiting the different profile in time of the signals. Among others, a typical parameter used in this analysis is the so-called pulse-shape-discrimination parameter (PSD), given by the ratio of the integrated charge in the tail of the signal with respect to the total integrated charge. An example is shown in Figure 1a, which shows how different particles (here thermal neutrons and cosmic muons) populate very different regions in the PSD *vs.* integrated-charge plane. For more details on the analysis and on the parameters used for the identification of the single events we refer to more specific studies (e.g., Cester et al., 2016).

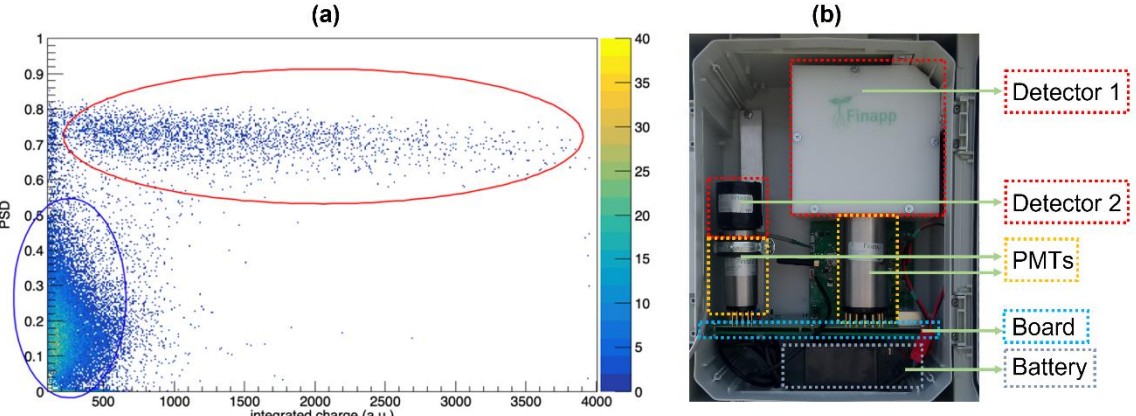

Figure 1. (a) Typical Pulse Shape Discrimination (PSD) *vs.* integrated charge plot for a FINAPP3 detector. Red and blue ovals indicate the neutron and muon region respectively; (b) scintillator-based sensor FINAPP3 with the two main detectors, photomultiplier (PMTs), board, and battery.

In the present study we use the scintillator-based sensor FINAPP3 developed by FINAPP.srl (finapptech.com/en). The main parts of the sensor are shown in Figure1b. The sensor hosts two main detectors. The first detector (Detector 1 in Figure 1b) is a multi-layer Zinc Sulfide Ag-doped scintillator mixed with Lithium-6 Fluoride powder

embedded in a silicone-based matrix. Epithermal neutrons are further moderated by the polyethylene shield and brought to thermal energies (around 0.026 eV) where neutron capture cross section on Li-6 is maximum. The Li-6 embedded inside the detectors has a large cross section for neutron capture. When a Li-6 nucleus captures a neutron, a nuclear reaction occurs and the compound Li-7 brakes into an alpha particle (He-4) and a triton (H-3) with a large energy release of almost 5 MeV. This energy is converted into light (a flash of optical photons) by the ZnS(Ag) crystals. The energy release in the thin layers of the scintillator (a few hundreds of microns) is strong for local interactions coming from the neutron-Li capture reaction products providing a large electrical signal, well above the voltage threshold used to cut the instrument noise. This detector can measure cosmic-ray induced muons too (in the energy of around 4 GeV) distinguished by a real-time PSD as described above. The possibility to detect muons in the same device was proven by the comparison with standard muon telescopes (patents n. IT102021000003728). The second main detector (Detector 2 in Figure1b) is a small (2" x 2") commercial organic scintillator (EJ200, from Eljen Technology Inc.). Due to the low effective atomic number $Z_{eff}$, typical of organic materials, gamma rays interact with this scintillator mainly by Compton scattering providing the spectrum shape of the Compton continuum from zero to the Compton edges. In the energy above 3.0 MeV no gammas are present but only signals with larger energy deposit (e.g., 10 MeV) due mainly to cosmic muons. For this reason, this second detector can measure muons as the first detector but also the total gamma rays fluxes in the energy range between 0.3 MeV and 3.0 MeV. For more details about the detected signals we refer to more specific studies (Boo et al., 2021; Ford et al., 2008). Finally, two commercial photomultipliers (PMTs in Figure1b), from Hamamatsu Photonics (Hamamatsu, Japan) are used to transform the light (visible photons) to electric pulse. The sensor can be further integrated with air pressure, air temperature and air humidity sensors. A single electronics board takes care of detector signal acquisition, real-time data processing and data logging to a remote server. All the components of the detector are in a box of about 40 x 30 x 20 cm with a total weight of 8 kg. Energy consumption is minimized to 0.4 Watt (35 mA at 12 V) and it is supplied by a relatively small solar panel (20 Watt) installed above the sensor. Overall, the new sensor assembly provide neutrons, muons and gamma counting rates that can be further corrected and elaborated to retrieve soil moisture as described in the next sections.

**2.2 From neutron counts to soil moisture estimation**

The measured neutron count rates $N$ are corrected for air pressure ($f_p$), variability of incoming neutron flux ($f_i$) and air vapour ($f_v$) to account for local atmospheric effects based on the following correction factors (Zreda et al., 2012):

$$f_p = exp\left(\beta\left(p - p_{ref}\right)\right) \tag{1}$$

$$f_i = \frac{I_{ref}}{I} \tag{2}$$

$$f_v = 1 + \alpha\left(h - h_{ref}\right) \tag{3}$$

$$N_c = N \cdot f_p \cdot f_i \cdot f_v \tag{4}$$

where, $\beta = 0.0076$ [mb$^{-1}$], $\alpha = 0.0054$ [m$^3$ g$^{-1}$], $p$ and $h$ are air pressure [mb] and absolute humidity [g·m$^{-3}$], $I$ is the incoming flux of cosmic-ray neutrons induced by galactic primary particles in the Earth's atmosphere [counts hour$^{-1}$, cph], $h_{ref}$, $p_{ref}$ and $I_{ref}$ are reference values (here the average is taken) of air pressure, absolute air humidity and incoming neutron flux during the measuring period, respectively. Air pressure and relative air humidity are

generally measured locally (or taken from a weather station nearby) and the latter can be converted into absolute air humidity using measured air temperature. In contrast, data of the incoming fluctuations are commonly downloaded (e.g. from https://www.nmdb.eu/nest/) from dedicated neutron incoming monitoring stations located at some places globally (Simpson, 2000). For the specific case study, data from JUNG station at Jungfraujoch
(Switzerland) are used for the correction as commonly adopted in many applications in central Europe (Bogena et al., 2022).

Finally, the corrected neutron count rate $N_c$ is transformed to volumetric soil moisture $\theta$ based on Desilets equation (Desilets et al., 2010):

$$\theta(N_c) = \left( \frac{0.0808}{\frac{N_c}{N_0} - 0.372} - 0.115 - \theta_{offset} \right) \cdot \frac{\rho_{bd}}{\rho_w} \tag{5}$$

where $\rho_{bd}$ and $\rho_w$ are the soil bulk density (kg·m$^{-3}$) and water density (kg·m$^{-3}$), respectively; $\theta_{offset}$ is the combined gravimetric water equivalent of additional hydrogen pools, i.e., lattice water (*LW*) and soil organic carbon (*SOC*), and $N_0$ is approximately the counting rate of the detector at a site during very dry soil conditions. The value $N_0$ can be calibrated based on independent soil sampling campaigns as suggested in different studies (Schrön et al., 2017; Franz et al., 2012). The data processing described above has been implemented in a simple spreadsheet available
from (Baroni, 2022b). For a more advanced data processing integrating also additional external data-sets readers can refer to Power et al. (2021).

### 2.3 Assessment of neutron counts to other conventional CRNS sensors at two sites (Austria and Germany)

The comparison to other conventional gas-tube-based CRNS detectors has been conducted at two experimental sites (Figure 2). The first site is located at Marchfeld (near Vienna, Austria, N48.24, E16.55). The second site is
located at Marquardt (near Potsdam, Germany, N52.45, E12.96). The recorded time series cover the period of seven months starting from May 2021 when, in both sites, a FINAPP3 detector was installed.

At Marchfeld experimental site, the FINAPP3 sensor is compared with a CRS2000, a boron-10 trifluoride proportional gas tube produced by Hydroinnova LLC (www.hydroinnova.com) that has been used in many studies (Andreasen et al., 2016; Baroni and Oswald, 2015; Hawdon et al., 2014). At the Marquardt site, several CRNS
sensors of different design are available for comparison (Heistermann et al., 2023). In the present study we selected a sensor based on two boron trifluoride proportional gas tubes (a double CRNS sensor system type called BF3-C-4) from "Lab-C" LLC, sold by Quaesta Instruments (www.quaestainstruments.com). This sensor provides a high sensitivity for neutron detection, thus good signal-to-noise ratio, which can promise potential for estimating soil moisture at even about hourly time resolution (Fersch et al., 2020).
All the detectors have been installed at a height of around 1.5 m above the ground and less than a few meters distance. Considering the large footprint of the signal detected, this horizontal difference is considered negligible for the comparison (Rivera Villarreyes et al., 2011; Patrignani et al., 2021; Schrön et al., 2018). All the detectors have been equipped with a solar panel and with GSM data transmission for supporting long-term observations and real-time monitoring.

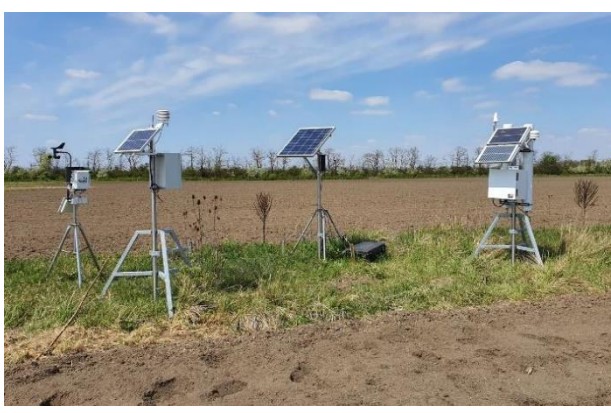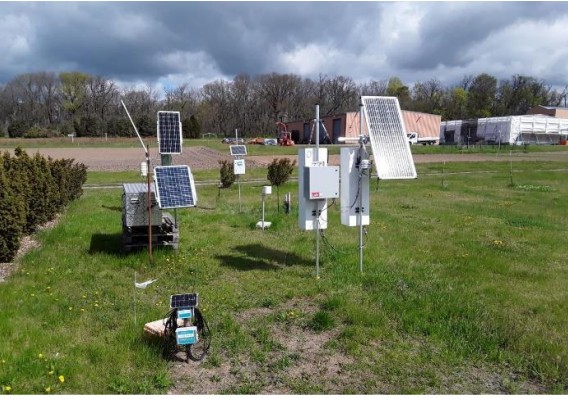


Figure 2. Experimental sites (left) Marchfeld (near Vienna, Austria) and (right) Marquardt (near Potsdam, Germany)

**2.4 Assessment of derived soil moisture with independent gravimetric soil sampling campaigns (Italy)**

A second assessment of the FINAPP3 sensor was carried out by a series of independent gravimetric soil sampling campaigns. The experiments were conducted in 2021 at four experimental sites located in the Po river plain,

northern Italy (Figure 3). At San Pietro Capofiume (N44.65, E11.64, near Bologna, Italy) and at Legnaro sites (N 45.34, E11.96, near Padova, Italy), the sensors were installed over a grassland with low biomass that is surrounded by agricultural cropped fields. Conversely, at Ceregnano (N45.05, E11.86, near Rovigo) and at Landriano (N45.31, E9.26, near Pavia, Italy), the sensors were installed in the middle of agricultural fields where fast biomass growth and irrigation took place. More specifically, at Landriano, sorghum was cropped and irrigated by a sprinkler

system. At Ceregnano, soybeans were cultivated and irrigated by a variable rate irrigation ranger system. The soil texture at the experimental sites is quite homogenous over the main area investigated by the sensors (approximately 100 m radius) except for Ceregnano, where a sandy fluvial deposit crosses the loamy field.

At each site, weather data were collected by meteorological stations operated by the Regional Environmental Protection Agencies (ARPA) at the same positions where the CRNS sensors were installed or located in close

distance (few km). In these cases, the meteorological observations have been considered representative for the local conditions. Moreover, three field campaigns were conducted during the vegetation season to collect soil samples for the calibration and assessment of the CRNS signal. The sampling took into account the sensitivity of the signal decreasing with distance from the sensor. Specifically, undisturbed soil samples were collected at 18 locations (red points in Figure 3) and at four different depths (0-5 cm, 10-15 cm, 20-25 cm and 30-35 cm from

the soil surface) for a total of 72 soil samples. Gravimetric water content for each soil sample was measured by dry-oven method (105° for 24 h). A mixed soil sample was further prepared at each site to measure soil organic carbon (SOC) and Lattice Water (LW). These two parameters have been measured by a Loss On Ignition (LOI) method respectively with a cycle of 24 h at 500° C and 12 h at 1000° C (Barbosa et al., 2021). All the values have been processed to account for the spatial sensitivity of the neutrons detected based on the most recent methods

(Schrön et al., 2017). A simple spreadsheet where these weighting functions have been implemented is publicly available (Baroni, 2022b). The results are summarized in Table 1 in the appendix.

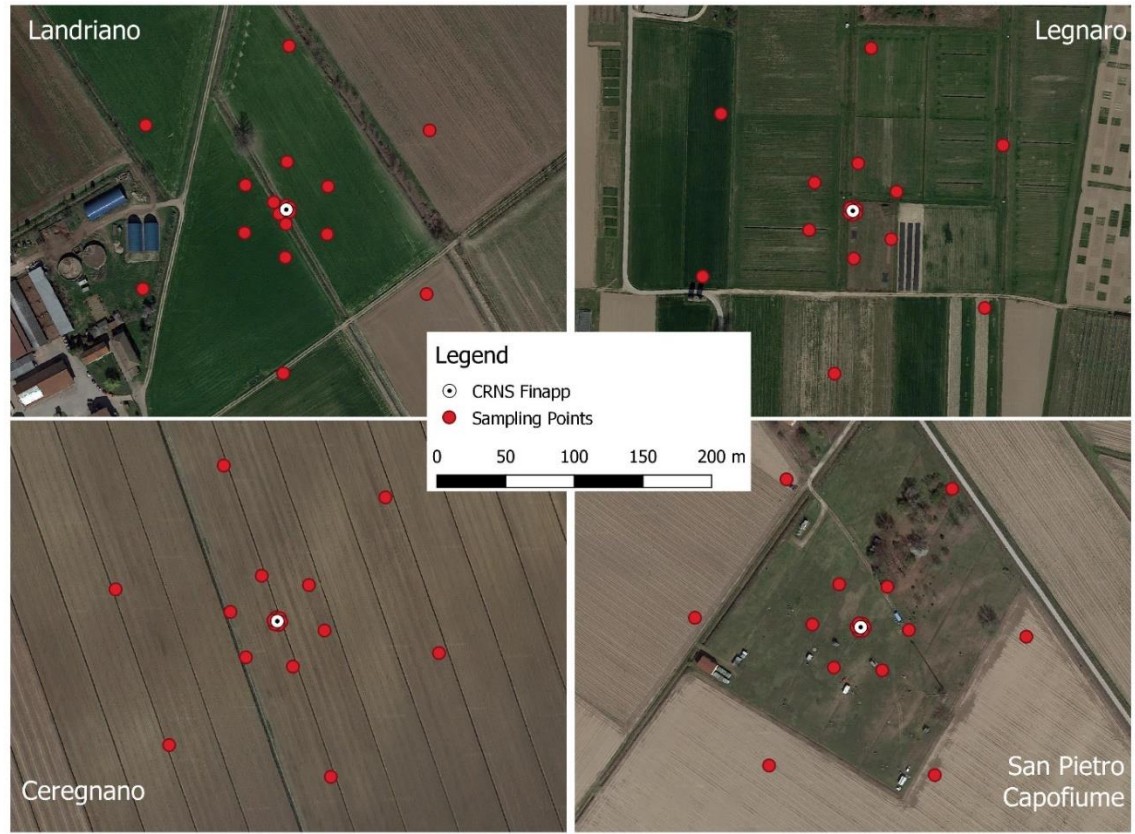

Figure 3. Experimental sites with FINAPP3 sensor (white points) and locations where gravimetric soil samples (red points) have been collected for comparison (pictures from Google Earth). At Ceregnano site, a gamma ray spectrometer Medusa Radiometrics gSMS was also installed few meters from the CRNS sensor.

### 2.4 Assessment of muons counting rate

The use of muons has been shown to be a possible alternative to the use of the neutron monitoring stations for incoming correction since they are produced from the same cascade as cosmic-ray induced neutrons in the atmosphere (Stevanato et al., 2022). We also test this signal in the present study and for sake of clarity we report here the main data-processing steps. Specifically, muons are first corrected to account for air pressure and air temperature effects as follows:

$$f_{p\_M} = exp\left(\beta_M\left(p - p_{ref}\right)\right) \tag{6}$$

$$f_{T\_M} = 1 - \alpha_M\left(T - T_{ref}\right) \tag{7}$$

$$M_c = M \cdot f_{p\_M} \cdot f_{T\_M} \tag{8}$$

where Eq. (6) is analogous to the pressure correction for neutron flux (see Eq. 1), $p$ and $T$ are the air pressure [mb] and air temperature [°C], respectively, and $p_{ref}$ and $T_{ref}$ are the reference value (here the average is taken) of air pressure and air temperature during the measuring period. In contrast to the neutrons, the effect of air vapour on muon counting rate has been not identified so far (Dorman, 2004; Maghrabi and Aldosary, 2018) and it is also not considered in the present study. Noteworthy, the whole air temperature profile should be considered for the correction. This would better represent the atmospheric condition and it would better capture the effect on muons. Some studies, however, have shown how the use of air temperature measured at 2 m hight provides a good approximation on the muon effect (de Mendonça et al., 2016). This approach is used also in this study, but it should be further tested in future research.

For the muon assessment, first the parameters $\beta_M$ and $\alpha_M$ are derived based on the data collected within this study to evaluate the effect of air pressure and air temperature on the muon signal. These values are then compared with $\beta_M = 0.0016$ mbar$^{-1}$ and $\alpha_M = 0.0021$ °C$^{-1}$ provided by (Stevanato et al., 2022). These values have been estimated based on a recursive analysis conducted on a relative long time series collected at the same area (one year time series collected at around 200 km distance). For this reason, the values can also be representative for the experimental sites of the present study. Refinements of these values should be expected in case of application in different locations. The corrected muon flux $M_C$ is then compared to the incoming variability measured at the neutron monitoring station usually adopted for CRNS incoming correction (https://www.nmdb.eu/). Finally, the effect of using muon signal instead of using neutron counts from a neutron monitoring station for the incoming correction (Eq. 2) and soil moisture estimation is also presented and discussed.

**2.5 Assessment of total gamma rays**

The measurements of gamma rays has been shown to be a valid approach for soil moisture estimation at relative small scale, i.e., tens of meters (Baldoncini et al., 2018) or for identifying irrigation events at agricultural sites (Serafini et al., 2021). More specifically, gamma-rays measured above the ground (e.g., by a detector installed about 2 meters from the ground) are mainly produced by radionuclides in the soil. The gamma-ray fluxes are also attenuated by the presence of water in the soil, due to the increased average absorption coefficient of the wet soil with respect to the dry soil. For this reason, the gamma-ray signal (i.e., the $^{40}$K full-energy peak at 1.46 MeV or, anyhow, in the energies between about 1.0 MeV to 2.5 MeV) shows a negative correlation with the amount of water in the soil and thus this relation can be used to estimate soil moisture dynamic (Strati et al., 2018). In contrast, gamma-rays in the energy range of $^{214}$Pb (352 keV), a radon progeny, has a much stronger volatility and it is also present in the atmosphere. Thus, a fast increase in the gamma-rays in the energy of this photopeak can be detected during precipitation events due to the effect of radon atmospheric deposition. In contrast, during an irrigation event, no such behaviour is expected. Noteworthy, the gamma signal should not be corrected for other effects (i.e., air pressure, air temperature and air humidity). For these reasons, it can provide some advantages to the use of neutrons for soil moisture application.

For the assessment of the gamma signal measured by FINAPP3, a stationary CsI gamma-ray spectrometer (gSMS, Medusa Radiometrics, https://medusa-online.com/en/) has also been installed at Ceregnao site in 2021, few meters from the CRNS location. A direct comparison between total gamma fluxes measured by the two sensors is performed. The capability of the signal to discriminate precipitation and irrigation events is also explored in the present study based on the data collected at the experimental sites.

**3. Results**

**3.1 Comparison between neutrons detected by FINAPP3 and conventional CRNS sensors**

The corrected hourly neutron count rates measured by the different sensors are shown in Figure 4. As expected, the sensors have different sensitivities with mean neutron counting rate over the period at Marchfeld of 1279 cph and 1797 cph, for FINAPP3 and CRS2000, respectively and at Marquardt of 1187 cph and 8387 cph, for FINAPP3 and Lab-C, respectively. Accordingly, the relative lower sensitivity of FINAPP3 produced a higher amount of statistical noise when compared to its benchmark (CRS2000 or Lab-C, respectively). However, this difference is less substantial when the signal is smoothed over a longer time interval. Specifically, the analysis shows a good

agreement of the detected signals ($R^2 = 0.66$) at 1 hour integration time. The performance improves ($R^2 = 0.91$) when the values are integrated already over six hours interval. The good correlation can also be appreciated by looking at a fast drop of the neutron counting rates during a short time scale (Figure 4c, d). For this reason, the

FINAPP3 sensor can be considered reliable for many applications while it is suggested to employ a more sensitive detector for especially demanding settings, e.g., when focusing on fast (e.g., hourly) hydrological processes like canopy interceptions (Andreasen et al., 2017a; Baroni and Oswald, 2015) or mobile applications (Jakobi et al., 2020).

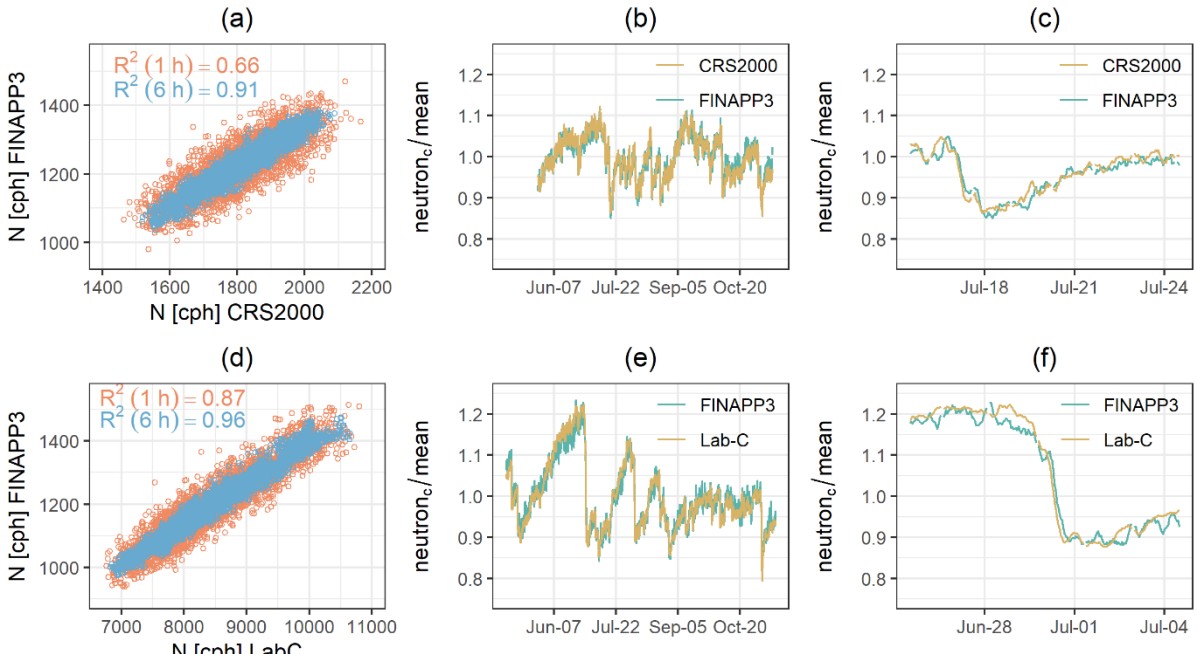

Figure 4. Comparison of measured neutrons in 2021 at Marchfeld site, Vienna, Austria (top row) and Marquardt site, Potsdam, Germany (bottom row) by the two different sensor pairs (CRS2000 and FINAPP3; Lab-C and FINAPP3). Plots (a) and (d) show the hourly values in orange and based on a running average of 6 hours in blue. Plots (b) and (e) show the neutron fluxes corrected for air pressure and with a running average of 6 hours. The relative counts over the mean are shown for comparison. Plot (c) and plot (f) show a zoom-in during a fast drop of the neutron counts.

**3.2 Assessment of the derived FINAPP soil moisture with independent gravimetric soil samples**

The neutron counts collected at the four Italian experimental sites were transformed to volumetric soil moisture as described in section 2.2 using all the soil samples for the calibration of the parameter $N_0$ (Eq. 5). Before the transformation, the corrected hourly neutron values were smoothed with a Savitzky-Golay filter to decrease the random fluctuations at short time period as suggested in literature (Franz et al., 2020). The calibration curves

obtained based on all the gravimetric soil samples are shown in Figure 5 (dashed black lines) together with some performance metrics between estimation and observation (coefficient of determination $R^2$ and RMSE). Moreover, calibration curves based on the data collected during only one single soil sampling campaign are added to better visualize the differences (grey lines).

At the Legnaro site, the calibration curve aligned well the observations with a high goodness of fit ($R^2 > 0.9$;

RMSE = 0.006 g g$^{-1}$). In contrast, at the other three sites, the goodness of fit deteriorated with the worst case obtained at Ceregnano site ($R^2 > 0.2$; RMSE = 0.041 g g$^{-1}$). These performances are in agreement with studies conducted with other conventional CRNS sensors (e.g., Franz et al., 2012) and they can be explained in relation i) to the effect of other hydrogen pools like biomass (Baatz et al., 2015; Franz et al., 2015; Jakobi et al., 2018) and ii) to the contributions to the signal from remote areas (Schattan et al., 2019; Schrön et al., 2017).

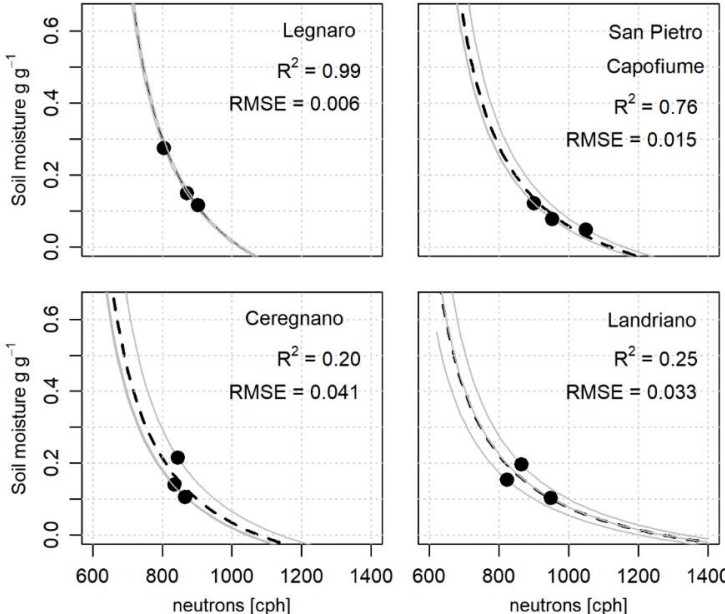

Figure 5. Calibration curves obtained at each site (Legnaro, San Pietro Capofiume, Ceregnano and Landriano) using data collected during one single field campaign (gray lines) or based on the best fit over all the samples (dashed black line).

Specifically, the very good fit at Legnaro site can be explained considering that the FINAPP3 sensor has been installed at a grass site with low biomass and the surrounding areas are characterized by relatively small

agricultural fields (see Figure 3). In these conditions, the soil samples well represent the average soil moisture within the footprint and no additional hydrogen pools are relevant. As such, the results support the sufficiency of one single calibration campaign and the accuracy of the detected signal when these conditions are met. At San Pietro Capofiume, the FINAPP3 sensor was also installed at a grass site with low biomass. This area, however, reached very low soil moisture values during the summer. In contrast, the remote areas are large, irrigated maize

cropped fields (i.e., with much higher expected soil moisture). As recently discussed (Schrön et al., 2023), in these particularly heterogeneous conditions, the sensor can detect soil moisture changes at more remote distance than the actual footprint and the gravimetric soil samples collected during the field campaigns could be not representative of the average soil moisture condition detected by the sensor. On the one hand, this can explain the unrealistic apparent negative soil moisture values estimated during August. On the other hand, it supports the need

of additional soil samples at the irrigated areas to provide a soil moisture basis more representative for this CRNS footprint. Finally, at Ceregnano and at Landriano, the FINAPP3 sensors were installed at the centre of a homogenous cultivated field where the contribution of the fast biomass growth to the detected signal should be expected. Thus, the apparent overestimation of soil moisture towards the peak of the growing season at both sites is very plausible. Some corrections to the signal to account for the biomass contribution have been suggested in

literature (Baatz et al., 2015; Franz et al., 2015; Jakobi et al., 2018) but it is beyond the aim of the present study to assess these approaches. The use of more recently proposed soil moisture-neutron relation could also be tested in future studies to see possible compensation for these effects (Köhli et al., 2021). Anyway, these results confirm the need to conduct when possible more than one calibration campaign to account for some of these effects (Heidbüchel et al., 2016; Iwema et al., 2015).

Finally, the time series collected at the four experimental sites in Italy are shown in Figure 6. The FINAPP3 signal was regularly recorded and transmitted over the entire period. Only few data gaps were experienced, and they are

related to short periods of low power supply by the solar panel during wintertime. At all the sites, the estimated soil moisture dynamic responds well to precipitation. As previously discussed, the derived soil moisture values are in good agreement with the gravimetric soil moisture (green crosses). For these reasons, the results show how FINAPP3 can be considered a reliable soil moisture sensor to be integrated in long-term monitoring networks, as proposed by (Cooper et al., 2021; Zreda et al., 2012; Bogena et al., 2022).

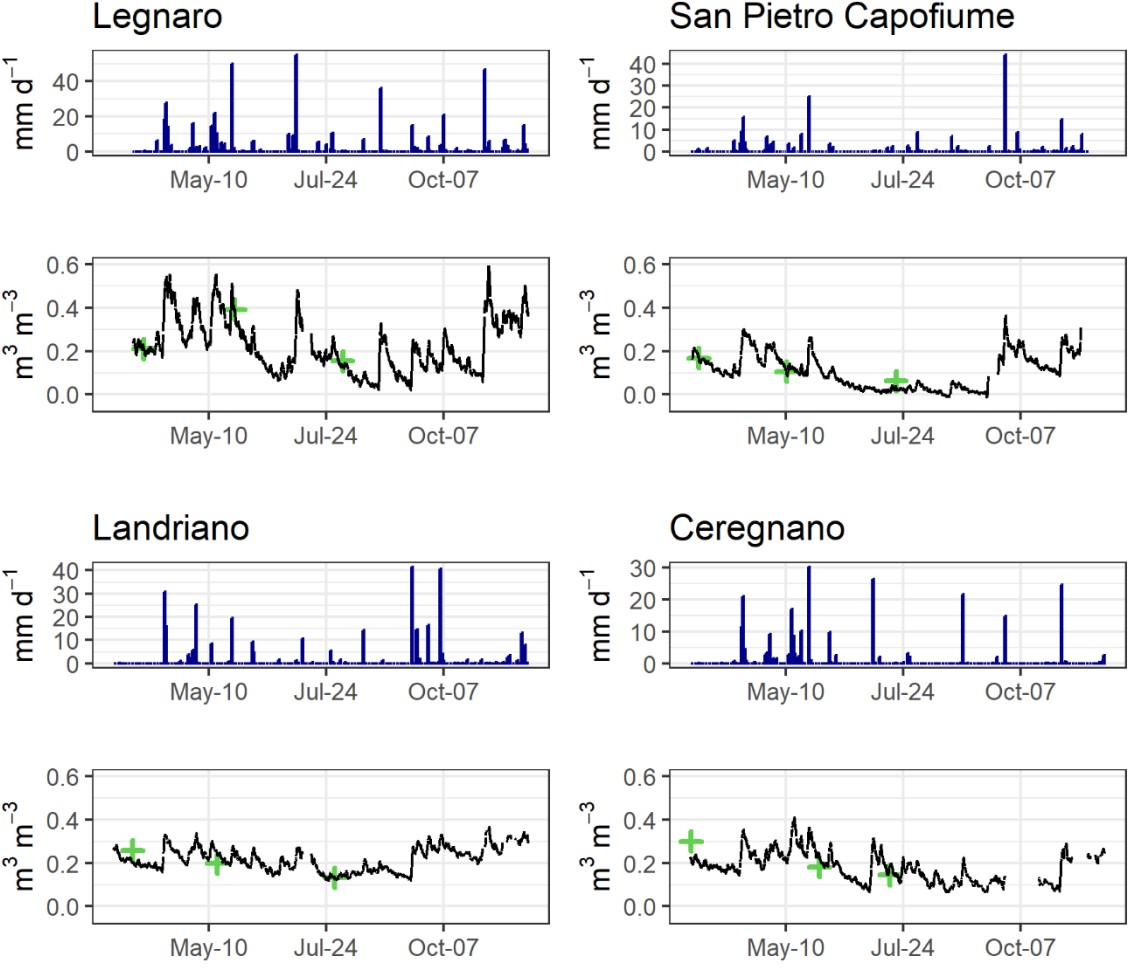

Figure 6. Estimated volumetric soil moisture ($m^3$ $m^{-3}$) by FINAPP3 in 2021 at the four experimental sites (black line) compared to weighted average soil moisture based on soil samples and gravimetric methods (green crosses). At each site, the precipitation is also shown (blue bars).

### 3.3 On the use of muons for incoming corrections

Muons have been recorded simultaneously by the detector at all the experimental sites. Some malfunctions in the pulse-shape-discrimination integrated in the electronic board and on the data transmission have been however initially identified. These malfunctions have been later fixed but some data have been corrupted. For this reason the muon time series cover a shorter period in comparison to the neutron counts (i.e., June – November). Figure 7 shows the moun counting rates collected at Legnaro site, as example, but similar results have been detected in the other experimental sites. As expected, the results show a strong relation between measured muon counting rates and air pressure (Figure 7a). The slope of the relation (-0.0018) is also very similar to the value obtained by Stevanato et al. (2022) (i.e., -0.0021). In contrast, within the present study no relation is detected between the pressure corrected muons and air temperature (Figure7b). The behaviour is attributed to the relative short time series and the small temperature range (±5°). However, the representativeness of air temperature measured at 2 m

hight in comparison to the need of a whole air temperature profile is also questionable and it should be further investigated (de Mendonça et al., 2016). The residual spread in the relationship suggests that the influence of factors to the signal other than cosmogenic muons cannot, however, be excluded and it should be considered in further studies.

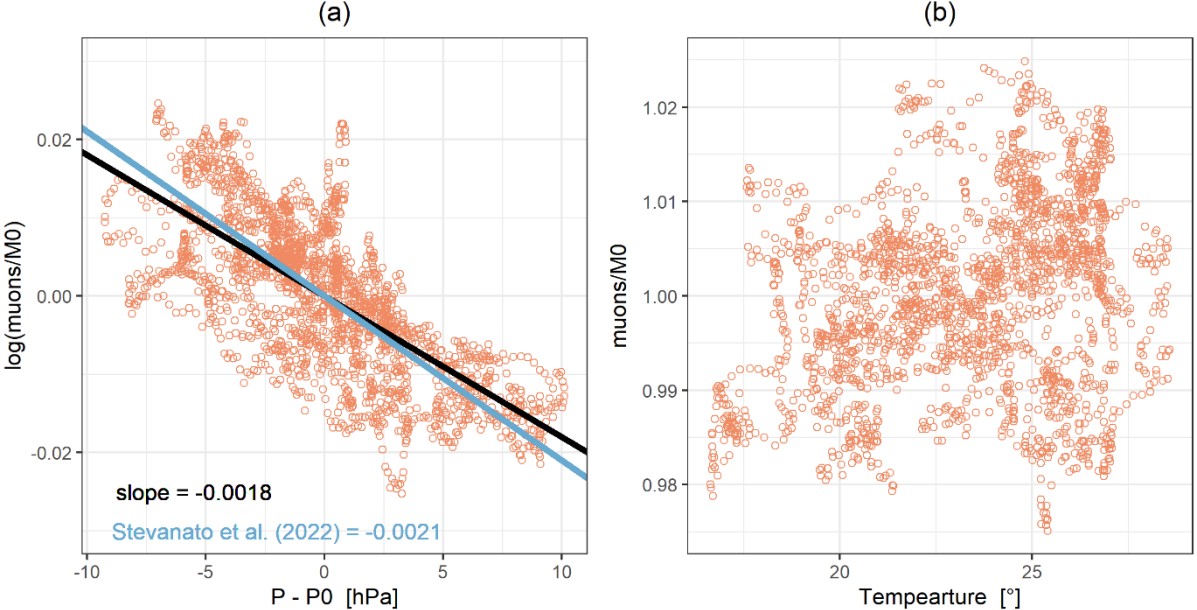

Figure 7. Comparison of data collected at Legnaro site: (a) relative air pressure *vs.* muon counting rate; (b) air temperature *vs.* corrected pressure muon counting rate.

The muon counting rate is further analysed by comparing its dynamic to incoming neutron fluxes measured at a neutron monitoring station (Jungfraujoch) and based on the effect on the derived soil moisture (Figure 8). During most of the monitoring period, the main fluctuations are clearly visible in both muon and incoming neutron (JUNG) time series (Figure 8, left). In some days (e.g., on 5th of July when a precipitation event occurred), some differences are detected that might be attributed to different local atmospheric conditions between the experimental sites and Jungfraujoch where the incoming neutron fluxes are measured. However, these differences do not propagate into significant differences in derived soil moisture. For this reason, the analysis within the present study is not conclusive but longer time series (e.g., years) with stronger incoming variability are needed to test the use of muons for incoming correction.

Noteworthy, one single relevant event has been recorded at the beginning of November (Figure 8, right). During this period, a fast drop in the incoming fluxes has been detected, producing ~8% increase in the incoming correction if neutron monitoring is concerned. In contrast, the fluctuations of the muons are much more smoothed. At the current stage, the reasons of these differences have been not identified but only some hypotheses are formulated. First, the FINAPP3 muons count rate is relatively low and the recorded signal is smoothed over relative long-time period (days) to reduce the statistical errors. For this reason, short term dynamics cannot be captured. Second, the muon detector is also not directional (e.g., as a telescope looking upward) but it measures muon particles that are scattered in all the directions. These characteristics could produce some differences in comparison to directional detector when these fast and strong events are considered. For this reason, the need of a bigger or directional muon detector could be considered for further developments to detect events that occurs during relatively short period. Still, it is interesting to note the propagation of these different corrections into soil moisture estimation. Specifically, a precipitation event was observed over all the Italian sites during this strong incoming

neutron variability. Accordingly, soil moisture should have increased to some degree. The effect of the incoming

correction based on the neutron monitoring station, however, smooth this effect and the soil moisture remains

constant or even started to dry down. In contrast, by using the muon signal, the soil moisture increases. While the

magnitude of this increment is in some cases questionable if compared for instance to the increment recorded

during the earlier precipitation event, these results support previous findings that muon detection can be a possible

approach to better account for local atmospheric conditions. In this context, FINAPP3 can be considered a valuable

sensor for collecting new data for further testing this hypothesis. The use of continuous independent soil moisture

measurements should be however designed for benchmarking.

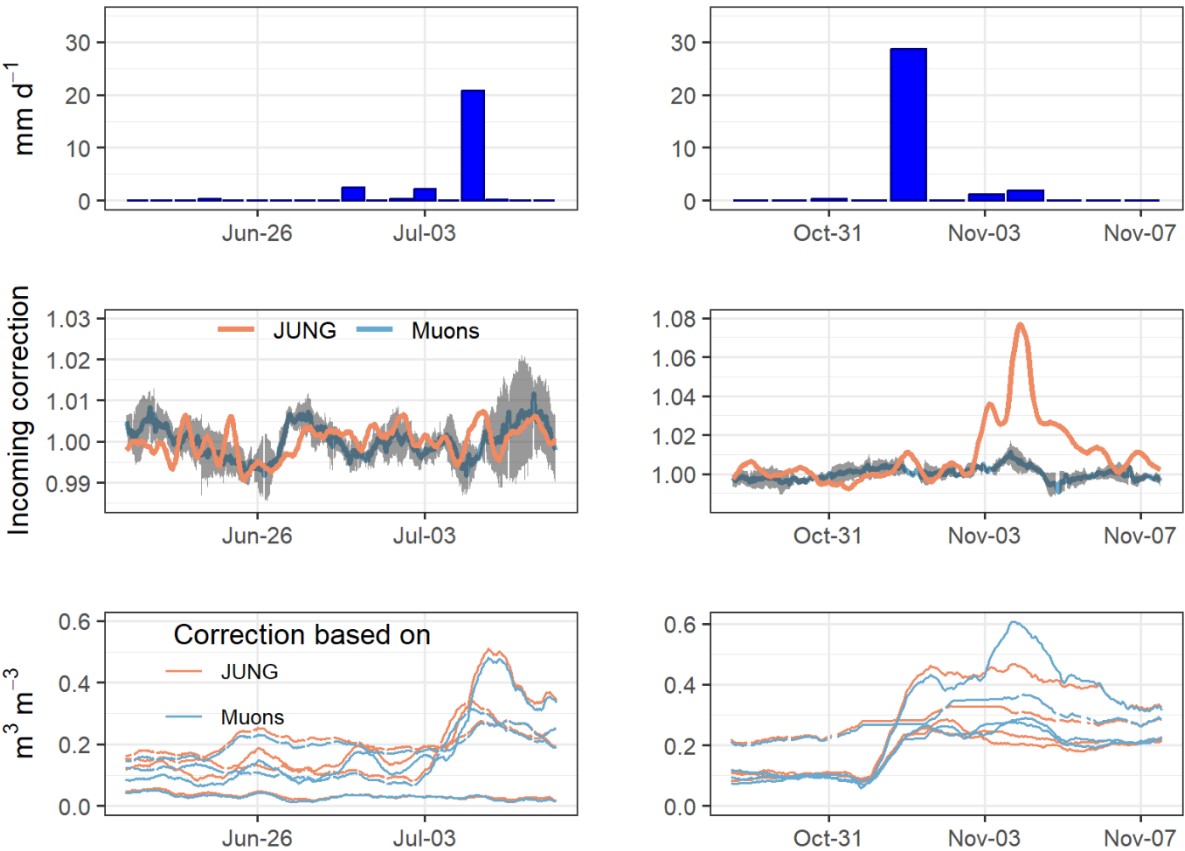

Figure 8. The plots in the top row show the average precipitation over the four Italian experimental sites. The plots at the middle
row show the incoming correction based on neutron monitoring station (JUNG) and based on the average muon detected at the
four experimental sites (Muons). Standard deviation is also shown as grey area. Bottom plots show estimated soil moisture
using the wo different approaches for the incoming correction of the signal: based on the standard approach of data from neutron
data base (e.g., JUNG plotted as orange line) and using locally detected muons (blue line). The measurements refer to the year
2021.

**3.4 Assessment of measured total gamma rays**

The comparison between total gamma counts (TGC) measured at Ceregnano site by FINAPP3 and gSMS Medusa

is shown in Figure 9. On average, the sensitivity of the FINAPP3 is lower with an average counting rate over the

monitored period of 2531 counts per hours (cph). In contrast the gSMS Medusa sensor showed higher sensitivity

and an average counting rate over the monitored period of 8281 cph. The correlation between the two signals is

low at 1 h time resolution ($R^2 = 0.08$), mainly due to the presence of extreme values observed during the

precipitation events. The correlation increases ($R^2 = 0.32$) with a consistent detected dynamic (Figure 8b) when

these extreme values are removed, and the time series is smoothed over a 6 h time window.

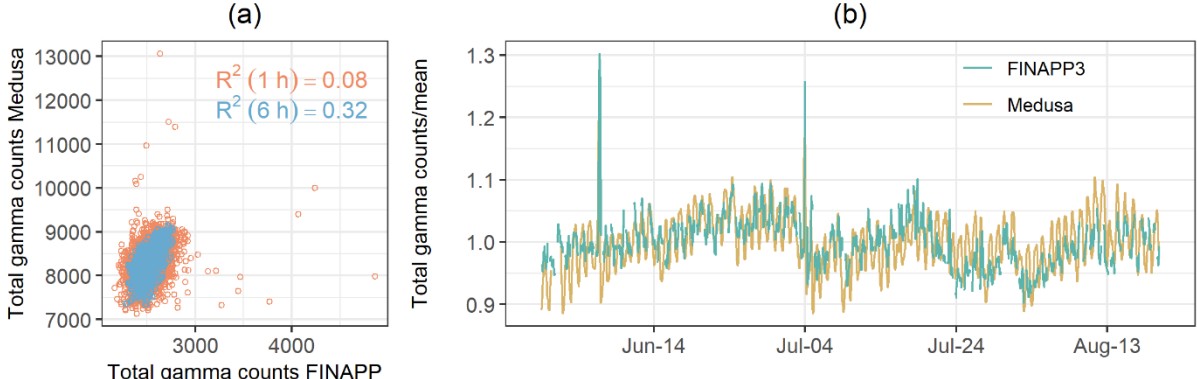

Figure 9. Comparison between total gamma counts measured by FINAPP3 and gSMS Medusa gamma-ray spectrometer on 2021 at Ceregnano site.

The measured total gamma counts are further compared to the soil moisture simultaneously derived by FINAPP3 and with precipitation and irrigation events (Figure 10). Please note that a relatively shorter time series (June – September) in comparison to the neutron time series is shown due to some malfunctions of the electronic board and data transmission that have been initially deprecated the gamma signal, as also discussed for the muon signal. The collected results show a negative correlation with the soil moisture dynamic estimated based on the neutron counts (i.e., TGC increases with soil moisture decreasing and *viceversa*). Thus, the results confirm how the total gamma fluxes are attenuated by the presence of water in the soil providing the scientific basis to develop a gamma-ray sensor for soil moisture estimation (Strati et al., 2018). However, the total gamma counts show higher dynamic at sub-daily time scale in comparison to the estimated neutron-based soil moisture and the correlation between the signals is weak (Pearson correlation coefficient r = -0.18). For this reason, further experiments and analyses should be conducted to better understand the added value of this signal for soil moisture estimation. Among others, the weak correlation can be attributed to the smaller horizonal and vertical footprint of the gamma fluxes (<25 m radius, <15 cm depth) in comparison to the neutron (~100 m radius, ~40 cm depth). Thus, a dedicated soil sampling campaign within the theoretical soil volume detected by the gamma particles should be performed for better assessment. An exponential decrease of the sensitivity of the signal has also been suggested in literature in both horizontal and vertical directions (Baldoncini et al., 2018). However, considering that the gamma footprint is strongly affected by the height of the detector installation (van der Veeke et al., 2021), further and more dedicated experiments should be performed to develop specific weighting functions and to conduct a proper assessment.

Noteworthy, however, a peak in the total gamma radiation generated by the deposition of atmospheric radon during the precipitation events is clearly identifiable. In contrast, no such peaks occur during the irrigation events. The results are shown in Figure 10 where two short periods are visualized as example. For this reason, while the use of total gamma radiation for soil moisture estimation will require additional refinements, the new sensor can be used for discriminating the increase of soil moisture due to irrigation in contrast to precipitation events as shown in other studies using more dedicated gamma-ray spectrometers (Serafini et al., 2021). The use of this signal to extend existing gamma ray dosimeters can also be foreseen (Rizzo et al., 2022).

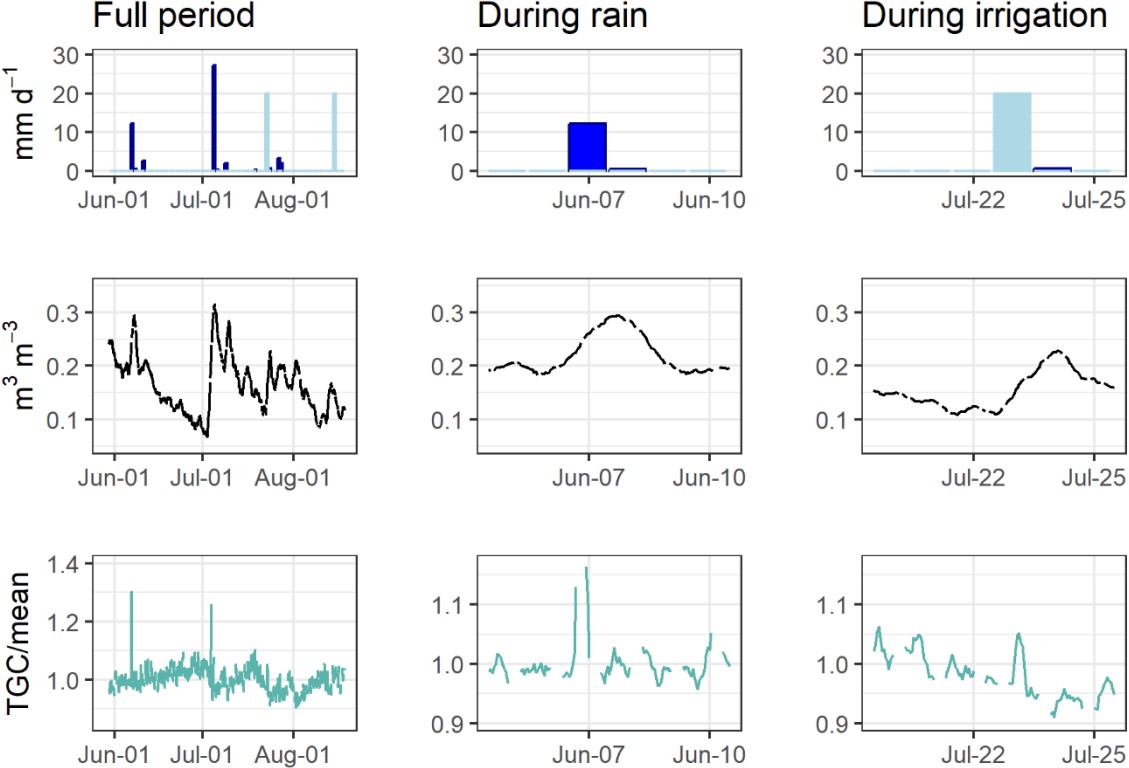

Figure 10. From top row, precipitation (blue) and irrigation (light blue) (mm d$^{-1}$), volumetric soil moisture estimated by FINAPP3 (m$^3$ m$^{-3}$) and total gamma counts (TGC) over the mean of the monitored period (year 2021).

### 4. Conclusions

This study presents the activities conducted to test a new CRNS sensor design based on scintillators for non-invasive soil moisture estimation. The results show that the new sensor performed very well in different environmental conditions in comparison to other conventional gas-tubes-based CRNS sensors (R$^2$ > 0.9 at 6 hours integration time) and based on several gravimetric soil moisture samples (RSME <0.04 m$^3$ m$^{-3}$). The sensitivity of this new sensor design was found suitable for monitoring daily temporal soil moisture changes over long term (years). However, the signal noise was relatively high at hourly time scale and only the aggregation to 6-h-interval yielded a reasonable robustness of the signal. For this reason, a more sensitive detector should be considered when fast hydrological processes such as canopy interceptions or roving applications are targeted.

Part of the tested sensor design are components that simultaneously measure muons and total gamma radiation. In previous studies, muons were found to be a potential candidate to support the correction for incoming cosmic rays (Stevanato et al., 2022). On the other hand, the use of gamma-ray spectrometry was identified as an alternative method for non-invasive soil moisture estimation (Baldoncini et al., 2018) and irrigation discrimination (Serafini et al., 2021).

The muons measured within the present study confirmed the negative correlation with the air pressure that has been found in literature (Stevanato et al., 2022; de Mendonça et al., 2016). The effect of the air temperature was however not identified suggesting the need of longer time series and a wider temperature range. The incoming correction using muons showed some differences to the incoming variability detected by the neutron monitoring station that could be attributed to different local atmospheric conditions. In most of the period, however, the effect on soil moisture estimation was negligible. Further analyses with longer time series should then be conducted to

better understand the added value of detecting this radiation form. The comparison also to other recently proposed alternatives like the use of neutron spectroscopy (Cirillo et al., 2021) or improvements on the use of neutron fluxes measured at the neutron monitoring station (McJannet and Desilets, 2023) should also be foreseen.

The sensor had also a good performance in the measurements of the total gamma radiation in comparison to a gamma-ray spectrometer ($R^2 = 0.29$, at 6 hours integration time). The signal also showed a negative correlation to soil moisture as presented in other studies with the focus on specific gamma energy ranges, e.g., $^{40}$K (Strati et al., 2018; Baldoncini et al., 2018). The correlation using total gamma counts is however weak (Pearson correlation coefficient r = -0.18) suggesting the need of additional studies for a better understanding of the signal response and of the footprint size for soil moisture estimation. In contrast, high peaks of total gamma radiation generated by shower of radon in the atmosphere have been detected allowing a clear identification of precipitation *vs.* irrigation events.

Overall, this tested sensor design show to be a valuable alternative to more traditional CRNS detectors for soil moisture estimation. Considering that it can be built smaller than conventional neutron systems and the potential benefit of the additional detection of muons and total gammas, it can also open the path to new and wider applications like space weather applications (Hands et al., 2021; Rizzo et al., 2022) and for monitoring agriculture water use (Foster et al., 2020).

## Appendix

Table 1. Results of the soil samples analyses at the different experimental sites. $\theta_a$ is the arithmetic gravimetric soil moisture; $\theta_w$ is the weighted average gravimetric soil moisture based on Schrön et al. (2017); $N_0$ is the calibrated parameter of the Eq. 5; $\theta_{bd}$ is the soil bulk density; *SOC* is the soil organic carbon and *LW* is the lattice water.

| Site | Date | $\theta_a$ [g/g] | $\theta_W$ [g/g] | $N_0$ [cph] | $\rho_{bd}$ [g/cm$^3$] | *SOC* [g/g] | *LW* [g/g] |
|---|---|---|---|---|---|---|---|
| San Pietro Capofiume | 15/03/2021 | 0.133 | 0.121 | 1468 | 1.384 | 0.014 | 0.084 |
| | 10/05/2021 | 0.098 | 0.077 | 1466 | 1.373 | - | - |
| | 19/07/2021 | 0.049 | 0.048 | 1540 | 1.295 | - | - |
| Legnaro | 29/03/2021 | 0.174 | 0.149 | 1565 | 1.409 | 0.022 | 0.152 |
| | 26/05/2021 | 0.247 | 0.275 | 1563 | 1.421 | - | - |
| | 03/08/2021 | 0.114 | 0.114 | 1578 | 1.336 | - | - |
| Landriano | 22/03/2021 | 0.210 | 0.196 | 1413 | 1.322 | 0.019 | 0.007 |
| | 15/05/2021 | 0.200 | 0.154 | 1274 | 1.285 | - | - |
| | 29/07/2021 | 0.125 | 0.103 | 1349 | 1.295 | - | - |
| Ceregnano | 10/03/2021 | 0.209 | 0.215 | 1501 | 1.397 | 0.018 | 0.076 |
| | 31/05/2021 | 0.178 | 0.140 | 1383 | 1.306 | - | - |
| | 15/07/2021 | 0.134 | 0.105 | 1376 | 1.386 | - | - |

**Code and data availability**

Data collected and processed at the six experimental sites are available at the following repository (Baroni, 2022a). Two spreadsheets have been developed for data processing. The first file (CRNS_SoS.xlsm) integrates the weighting functions for processing soil samples. The second file (CRNS_PoP.xlsm) integrates the atmospheric corrections and the calibration function to transform measured row neutrons to soil moisture. The spreadsheets can be downloaded at the following repository (Baroni, 2022b).

## Author contributions

Conceptualization GB, LS, SG, design and implementation of field experiments, methodology GB, LS, SG, TF, HA, AT, GW; writing - original draft preparation SG, GB. Writing - review and editing: all the co-authors. All authors have read and agreed to the published version of the manuscript.

## Competing interests

Luca Stevanato, Matteo Polo and Marcello Lunardon are of FINAPP S.r.l., 35036 Montegrotto Terme, Italy. Otherwise, the authors declare no competing interests.

## Acknowledgments

The study was partially conducted within the CRP IAEA project D12014 Enhancing agricultural resilience and water security using Cosmic-Ray Neutron Sensor and within the project 21GRD08 SoMMet that has received funding from the European Partnership on Metrology, co-financed from the European Union's Horizon Europe Research and Innovation Programme and by the Participating States. We also acknowledge the support of the colleagues of the regional environmental agencies (ARPAe, ARPA Lombardia and ARPA Veneto), Veneto Agricoltura and the Department of Agricultural and Environmental Sciences, University of Milan (Italy) for conveying the experimental sites and for the discussion of the results during the research activities.

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
