# Peer review of "Testing a novel sensor design to jointly measure cosmic-ray neutrons, muons and gamma rays for non-invasive soil moisture estimation"

_Geoscientific Instrumentation, Methods and Data Systems, 2022_

## Author Comment (AC1)

Testing a novel sensor design to jointly measure cosmic-ray neutrons, muons and gamma rays for non-invasive soil moisture estimation by Gianessi et al. https://doi.org/10.5194/gi-2022-20

**Author Response to Reviewer #1**

RC: *Reviewer Comment*, *AR: Author Response*

**RC: The authors present a novel set of experiments using a new sensor to simultaneously measure neutrons, muons, and gammas. The new sensor is compared to conventional sensors with satisfactory results for longer time periods of integration (i.e. 6 hrs for neutrons). The technology is lighter and potentially a lower cost, which will open up doors for more applications in science and in practical applications. The manuscript is well written and appropriate for the journal. I have a few suggestions that should be addressed prior to publication. Some instances of English grammar and word choice will need to be addressed.**

*AR: Thank you for the very quick and very positive feedback. All the comments and suggestions will be implemented in the new version of the manuscript as described in the point-by-point response below.*

**RC: L33: "Runoff generation"**

*AR: Thanks, it will be corrected in the new version of the manuscript*

**RC: L 40: "More recently, attention"**

*AR: Thanks, it will be corrected in the new version of the manuscript*

**RC: L 44: "has shown"**

*AR: Thanks, it will be corrected in the new version of the manuscript*

**RC: Figure 1: Are events outside of the blue and red ovals not included in the analysis?**

*AR: Figure 1 is only descriptive. The PSD vs. integrated charge plane is the simplest way to show where most of the neutrons and muons lie. In contrast, the actual selection of the events is based on the analysis of statistical outliers over different parameters. This information will be added into the new version of the manuscript.*

**RC: L165: For the gammas are there any corrections needed for pressure or air temperature/humidity variations?**

*AR: Based on current literature (Baldoncini et al., 2018, Serafini et al., 2021), gammas do not require corrections for pressure and air temperature and humidity. We agree that this has some advantages in comparison to neutrons and we will add this information in the new version of the manuscript.*

**RC: L247: I would use the SG filter on the neutron/gamma count time series, not the soil moisture time series that have been transformed by the calibration function.**

*AR: Thank you for the suggestion. We did not see relevant differences when applying the filter to the neutron/gamma time series or to the estimated soil moisture. But we acknowledge that the filter could be applied directly to the raw data. We will integrate this information into the new version of the manuscript.*

**RC: L 302-305: This sentence is confusing and long. Please rewrite.**

*AR: We have received from the Reviewer #2 and for Daniel Rasche (community comment) several comments and suggestions to improve the discussion about the use of muons for CRNS soil moisture estimation. Based on that, we will add in the new version of the manuscript much more details on the muon behavior and a comparison with neutron monitoring data base. The discussion of the result and this sentence will be rephrased accordingly.*

**RC: Figure 7. Will be interesting to compare the Muon detection with the correction factor being proposed by McJannet and Desilets using cutoff rigidity and atmospheric depth with the NMDB historical data. Unfortunately, that work is in the review process still.**

*AR: Thank you for sharing this ongoing study. We have searched for this publication, and we have seen that the paper has just now been accepted. Indeed, a comparison between that new proposed correction approach and the use of muons would be very interesting. As discussed also in the response to Reviewer #2, our current time series, however, are relatively short and do not capture strong incoming variability. For this reason, we consider this comparison beyond the scope of the present study. In contrast, we will stress that this new sensor design can provide in a longer term relevant data that can be used for testing different incoming correction approaches. We will integrate this discussion into the new version of the manuscript, and we will add the reference of McJannet and Desilets.*

**RC: Figure 8. So the soil moisture data is from FINAPP and not depth weighted following the Schron method? If you did have gravimetric surveys how would you depth weight them for the gamma sensitivity? From my understanding they would have a similar sensitivity with depth as the neutrons but be a little shallower (10-15cm?)?**

AR: *Yes, the Reviewer is right: soil moisture data is from the neutrons from Finapp. This estimated soil moisture is compared to total gammas. We are not aware of analytic functions that have been published to weigh point-scale soil moisture data to correspond to gamma signal. However, current literature suggests exponential decreases in both vertical and horizonal direction* (Baldoncini et al., 2018)*. So, we agree that gamma should have a similar sensitivity to neutrons but shallower depth and smaller footprint. We will add this information into the new manuscript to clarify how to better compare gamma signals to soil moisture series in future studies.*

**RC: Table 1. Iwema et al. 2015 recommends 3 calibrations for estimating N0. From the variability here I would do at least 3 to estimate some uncertainty on N0. I agree additional gravimetric studies are needed, particularly for establishing the gamma to soil moisture**

**dependence, especially when used in cropping systems with significant temporal variations in vegetation biomass. Unfortunately, for CRNS and GRS studies all roads don't lead to Rome but to more gravimetric sampling :)**

**Iwema, J., Rosolem, R., Baatz, R., Wagener, T., & Bogena, H. (2015). Investigating temporal field sampling strategies for site-specific calibration of three soil moisture–neutron intensity parameterisation methods. HESS, 19, 3203–3216. https://doi.org/10.5194/hess-19-3203-2015**

AR: *Thank you for the comment and for the reference. We agree that soil surveys still play an important role in assessing calibration functions and validating modelling results. We will integrate this comment and this reference in the new version of the manuscript.*

**References**

Baldoncini, M., Albéri, M., Bottardi, C., Chiarelli, E., Raptis, K.G.C., Strati, V., Mantovani, F., 2018. Investigating the potentialities of Monte Carlo simulation for assessing soil water content via proximal gamma-ray spectroscopy. Journal of Environmental Radioactivity 192, 105–116. https://doi.org/10.1016/j.jenvrad.2018.06.001

---

## Author Comment (AC2)

Testing a novel sensor design to jointly measure cosmic-ray neutrons, muons and gamma rays for non-invasive soil moisture estimation by Gianessi et al. https://doi.org/10.5194/gi-2022-20

**Author Response to Community Comment #1**

CC: *Community Comment*, *AR: Author Response*

**CC1: Dear authors,**

**I think that combining observations of neutrons for soil moisture estimation, gamma rays for distinguishing rainfall and irrigation as well as muons for correcting neutron observations as suggested by Stevanato et al. (2022) in a single sensor system is very interesting and would be a great advantage from a science and application perspective.**

**Cosmic-ray neutrons of different energies (e.g. Hubert et al. 2019) as well as muons (e.g. Braun et al. 2009) respond to solar events. As your observation period covers such an event, it poses an excellent opportunity to evaluate the muon data and their use for the correction of epithermal neutron observations. Thus, I suggest to add a time series plot covering the solar event (+/- a few weeks) showing the neutron monitor time series of the closest neutron monitors Jungfraujoch (JUNG) and Athens (ATHN) and the muon data of the four observation sites. A similar response to the solar event would underline the suitability of the sensor's muon product and the suggested correction approach.**

**Kind regards,**

**Daniel Rasche**

*AR: Dear Daniel Rasche, thank you for the comments and for the references. Based on your suggestions and from the comments of Reviewer #2, we have analyzed in more detail the dynamic recorded by neutron monitors (NMDBs) and the muons during our experiments. We present below a more comprehensive description of the results that we will integrate into the new version of the manuscript.*

*As shown in Figure 1 (top), the dynamics of incoming neutrons from NMDB (e.g., Jungfraujoch) and muons are generally well in agreement. We highlight however that some differences between the dynamic of NMDB and muons are identified (e.g., on 16th of July). These differences could be due to differences in local conditions, but this behavior is still under study. Noteworthy, the variability in incoming neutron and muon fluxes is in most of the period low and the differences between the two correction approaches do not significantly affect soil moisture dynamics computed therewith (e.g., Figure 1 bottom). For this reason, additional studies with longer time series and at different locations should be performed to test the use of muons for CRNS soil moisture corrections and to better understand these differences.*

[Figure]

[Figure]

*Figure 1. (top) Incoming correction factor based on neutron monitoring data base Jungfraujoch (Jung) and based on muon measured locally (Muon); (Bottom) effect of the different incoming corrections on estimated soil moisture at Ceregnano site.*

*The dynamics during the solar event on 4ᵗʰ of November is shown in more detail in Figure 2 below: from top, precipitation, muons and incoming neutron and muon fluxes, derived soil moisture at Ceregnano site, as example. As it is possible to see, in contrast to the general good agreement presented in Figure 1, the muon did not detect the strong and relatively fast drop in the incoming neutron fluxes measured by the NMDBs. At the current stage we cannot conclude about the reasons and we can only formulate some hypotheses.*

*On the one hand, we have to acknowledge that muon detector currently integrated in the Finapp3 sensor is not optimized for detecting fast changes: i.e., the detector is not directional (e.g., as a telescope looking upward) and instead it measures muon particles scattered from different directions, the counting rate is relatively low, and, also for this reason, the signal is processed smoothing the signal over days. The use of a bigger and directional detector for muon measurements to detect this relatively fast signal is, also for this reason, under investigation.*

*On the other hand, it is interesting to note the effect of the different corrections on the estimated soil moisture. As shown in the Figure 2 below, the solar event occurred during a precipitation event. As such, we should expect an increase in soil moisture. However, the soil moisture obtained using the incoming neutron fluxes for correction smoothed the signal. In contrast, the*

*soil moisture based on the muon shows a soil moisture increase. Also this hypothesis will be investigated in future studies with longer time series and possibly more solar events.*

*Overall, we want to underline also here that it was not the aim of the present study to conclude about the use of muons for CRNS soil moisture correction. In contrast, as discussed also in literature, this approach is at the early stage and only additional data will support this hypothesis and further developments. As also suggested by the Reviewer #2, sensors like the one presented in this study provides an excellent base to collect these new data. We will rephrase any statements that could be misleading, the figure and this discussion will be integrated into the new version of the manuscript.*

[Figure]

*Figure 2: from top, precipitation, incoming correction, estimated soil moisture based on the different incoming correction at Ceregnano site, as example.*

---

## Author Comment (AC3)

Testing a novel sensor design to jointly measure cosmic-ray neutrons, muons and gamma rays for non-invasive soil moisture estimation by Gianessi et al. https://doi.org/10.5194/gi-2022-20

**Author Response to Reviewer #2**

RC: *Reviewer Comment*, AR: *Author Response*

**RC: The study presents a new sensor system to monitor epithermal neutrons, cosmic-ray muons, and total (i.e. non-spectrometric) gamma rays all in one device. The novelty of this work is not only the individual technology of the three components, it is rather the combination of them and its potential for research and applications. Hence, it is important to demonstrate that each of the three signals can be reliably and simultaneously measured, and how they can be used for the greater good.**

**The neutron measurements seem to correlate well with traditional neutron detector signals, at least the scale of several days to months. Although the idea of correcting incoming neutron radiation with local muon measurements is not yet established and still an active field of research, the presented sensor may provide a fantastic opportunity to collect a large data set that could help to falsify this hypothesis in the future. The rather high RMSE of soil moisture calibration data and neutron products (up to 10 m3/m3 , Fig 5) is not a big issue, since every neutron probe would probably see similar deviation considering the substatial spatial heterogeneity, biomass, etc. I highly appreciate the elaborate discussion of influencing factors led by the authors.**

**A major weakness of the study is that no muon data was shown and comparisons to existing muon and gamma data sets are missing. This hinders proper falsification of these two measurement approaches. Many claims made in this study require better rigorous support. If possible, I'd suggest to add data to the study that provides evidence of the proper functioning of all three detectors, since it is key to covince the readers about the reliable measurement of three different particles at once.**

**The work is highly relevant to the journal and will have a great impact in the science community. However, I suggest major revisions to better streamline the focus of the paper on falsification and evaluation of the three components, of which muons and gammas are yet insufficiently addressed. The detailed comments below could help to quickly address the missing pieces before publication.**

*AC: We thank the Reviewer for the overall positive feedback. On the one hand, the Reviewer highlighted some weaknesses in the study. On the other hand, he/she also gave us very clear details on how these limitations could be addressed. We thank the Reviewer who appreciated the comparison and discussion about neutrons, and we agree that, in comparison, the assessment of gammas and muons have been less addressed. As discussed in more detail in the point-by-point response below, we will improve the assessment of the gammas by showing a comparison between data collected by the Finapp sensor and by a gamma-ray spectrometer detector. For the muons, we will show the collected data and we will compare the dynamic to the neutron fluxes measured at a neutron monitoring station (e.g., Jungfraujoch), as also suggested in the Community Comment by*

*Daniel Rasche (for details please also see Authors response to his comments). We will also extend the description and the discussion on the use of muons for soil moisture correction. We will better highlight that the use of muon for CRNS correction will be not fully addressed within this study but, as suggested by the Reviewer, rather that this sensor can provide data to test this hypothesis in future studies. Overall, we will rephrase the sentences to be more rigorous in the assessment. We think that the manuscript will be strongly improved based on all these suggestions and changes and we are looking forward to further feedback.*

**Major concerns**

**RC: The authors present a new device which combines three existing measurement principles, but only evaluate one of them with conventional devices (here, only neutron counter vs. other CRNS probes). Since the main novelty is the availability of three detectors at the same time, I would have expected also a validation of (or at least evidence for) the proper functioning of the muon and gamma detectors. This could be easily done with existing gamma ray probes from national authorities, and muon telescopes or the global muon network. Also consider plotting measured muon time series together with Jungfraujoch data to identify differences in their response to cosmic rays.**

*AC: We agree with the Reviewer, and we thank you for the suggestion to add additional data for comparison. We believe, however, that a proper comparison can only be performed when the sensors are collocated (installed at the same place) to avoid spatial differences in the signal driven by (among others) soil moisture and local atmospheric conditions.*

*For the gammas, we have data collected by a standard gamma-ray spectrometer ([https://medusa-online.com/en/](https://medusa-online.com/en/)) installed at Ceregnano site. As it is shown below (Figure 1 a), the correlation is low at 1 h resolution, mainly due to the presence of some outliers. The correlation increases at 6 h resolution and the dynamic is well captured (Figure 1b). We will add information on the gamma ray spectrometer and this comparison will be integrated into the new version of the manuscript.*

[Figure]

*Figure 1. Comparison between total gamma counts measured by Finapp3 and Medusa gamma-ray spectrometer.*

*The use of the gamma signal to discriminate precipitation and irrigation will be better visualized based on the Figure 2 below: from top, precipitation and irrigation, soil moisture by CRNS Finapp, gammas total counts. The left panels show the increase of total gamma counts during a precipitation event. In contrast (right panels), no such increase has been detected with the irrigation. These Figure 1&2 will replace Figure 8 of the first version of the manuscript.*

[Figure]

*Figure 2: from top, precipitation (left) and irrigation (right), soil moisture and total gamma counts.*

*Regarding muon detection in the Finapp probe, the possibility to detect muons in the same device which detects neutrons is the subject of one of the Finapp's Italian patents (no. IT102021000003728). This was proven by the comparison with standard muon telescopes (the Finapp probe was used both as single muon detector and as part of a telescope, providing the same identification output results as far as muons are concerned). This reference will be added in the new version of the manuscript.*

*In contrast, for the present study, we do not have muons measured independently for comparison. Still, as suggested by the Reviewer and by the Daniel Rasche in his community comment, we provide*

better evidence of the muons behavior and a direct comparison to incoming neutrons fluxes from neutron monitoring station (e.g., Jungfraujoch station). Based on that, we also better discuss the effect on the CRNS correction.

Specifically, to better show the proper functioning of the muon detector, Figure 3a below shows the correlation between the relative muon vs. atmospheric pressure at Ceregnano site (but similar results are obtained at the other sites – data not shown). The behaviour shows the inverse correlation that can be removed with an exponential factor (see eq.6 in the present manuscript). Please also note that the slope of this relation is very similar to the value obtained by Stevanato et al. (-0.0021). On the figure 2b below, the relation between muons and air temperature is shown. In this case, however, the correlation is low and it is not possibile to identify a clear relation. The main reason is attributed to the relative short time series and the small temperature range (±5 degree). For this reason, for the current study, we applied the parameters for the muon correction obtained based on the longer experiment conducted nearby presented and discussed in (Stevanato et al., 2022). The site is located at around 200 km distance from our experimental sites and it has similar characteristics (low land). For these reasons the parameters are considered valid. Still, the representativness of air temperature at 2 m hight is also still under study (see (de Mendonça et al., 2016)).

[Figure]

*Figure 2. (a) relative pressure vs muon ; (b) Air temperature vs muons.*

Finally, Figure 3 below shows the comparison between the incoming neutron flux measured by the neutron monitoring station Jungfraujoch and the Finapp probe muon measurement. The derived soil moisture is also plotted. As it is possible to see, the main fluctuations are clearly visible in both time series. In some periods (e.g., on 16th July), however, a deviation is detected. These differences could be related to the different locations where the detectors are placed. This hypothesis is still under study. However, since the variability is low, these differences in incoming correction do not propagate into significant soil moisture changes. For this reason, the collected data do not support further investigation and we acknowledge the need of longer time series with stronger incoming variability (e.g., years).

[Figure]

[Figure]

*Figure 3: (top) Incoming correction factor based on neutron monitoring data base Jungfraujoch (Jung) and based on muon measured locally (Muon); (Bottom) effect of the different incoming corrections on estimated soil moisture at Ceregnano site.*

*The main relevant incoming neutron variability is detected on 4th of November. During this period, a fast drop in the incoming fluxes has been detected, producing a 10% increase in the incoming correction. In contrast, the muons have not been affected (see Figure 4 below). At the current stage, we cannot conclude the reasons of these differences but only some hypotheses are formulated.*

*First of all, currently, the Finapp muon detector has been optimized to follow relative long term variability (weeks to months). The muons count rate is relatively low and we smooth the signal over relative longer time period to remove noise . The muon detector currently integrated in the Finapp probe is also not directional (e.g., as a telescope looking upward) but it measures muon particles that are scattered in all directions. This could produce some differences in comparison to directional detector when these fast and strong events are considered. For this reason, it is under investigation the need of a bigger muon detector to detect such an event that occurs in relatively short period.*

*Still, it is interesting to note the propagation of these corrections into soil moisture. Specifically, as discussed in the first version of the manuscript and also reported in the figure 4 below, the drop of*

*the incoming fluxes occurred during a precipitation event that affect most of the Italian sites, even if with different degree. Reasonably, soil moisture should have increased based on that precipitation event. The effect of the incoming correction based on the neutron monitoring station, however, smooth this effect and the soil moisture remains constant or even started to dry down. In contrast, by using the muon signal, the soil moisture is not dumped and rather increases. We fully acknowledge that at the current stage we cannot argue if the correction with muons improves soil moisture estimation or not. In contrast, we believe that the use of muons for CRNS soil moisture correction is at the early stage and this hypothesis should be tested in future studies with, e.g., time series of independent soil moisture measurements. For this reason, we agree with the Reviewer to better highlight in the new version of the manuscript that this type of sensor can provide an excellent data-sets to further investigate this behavior.*

[Figure]

*Figure 4. From top: precipitation; incoming variability based on muons measured at the four Italian sites and based on incoming neutrons measured at Jungfraujoch; derived soil moisture at Ceregnano site based on muons and Jungfraujoch incoming neutron correction.*

**RC: Muon processing is a bit vague. Why is there no correction for atmospheric humidity on the muon signal? Where do you get the temperature data from? Local near-surface temperature is probably not suited to represent temperature effects in the upper atmosphere. The authors also mention that long-term local time series have been used to estimate the parameters, but the data is not shown and as they cite Stevanato et al, it is not clear whether this is valid for both, the German and the Italian sites.**

*AC: We will better describe the muon processing as discussed below.*

*As reported in literature* (de Mendonça et al., 2016)*, muons are affected by pressure and air temperature. The effect of relative humidity has been investigated less, due to the absence of a relationship between the two variables as suggested by some researchers* (Dorman, 2004; Maghrabi and Aldosary, 2018)*. However, we cannot exclude this effect and we will also add this statement in the new version of the manuscript. The pressure can be measured locally and used for correction. In contrast, we agree that temperature measured at the high atmosphere should be used to account for this effect. Some studies found however that also local near surface temperature can be a good proxy* (de Mendonça et al., 2016)*. We are currently using this approach that has shown to be valid within the requested precision for the soil moisture application. We acknowledge however that further comparisons are necessary to provide better quantification of the related uncertainty.*

*Finally, in the present study we use the parameters for pressure and temperature correction that has been derived and presented by* (Stevanato et al., 2022)*. The parameters have been extracted by the analysis of the one-year muon data series taken in Padova (Italy), from November 2019 to October 2020. In contrast, the data-sets collected and presented in the present study are too small to establish site-specific relations (see also comments above). Moreover, the stations are within a distance of around 200 km and at similar altitude. For these reasons, we assumed in the present study that the parameters are valid. In contrast, we did not use muons correction for Germany and Austria sites where we agree that differences could be relevant and locally adjusted parameters would be preferrable. This discussion will be added in the new version of the manuscript.*

**RC: The authors present excellent agreement of their neutron data with conventional neutron detectors during 6 months. However, it is the short-scale effects which are particularly interesting, e.g. the coincident response to the onset of rain events, to dew formation, to atmospheric variations, to drying periods, etc. They could be different if detection energy or geometry differs, and could eventually shed light on the actual sensor performance. This is not identifiable from the presented data, at least not with the long x-scale chosen in Fig 4. I'd appreciate if the authors could provide a zoom-in of the data that allows for day-to-day analysis and comparison.**

*AC: for Marchfeld and Marquardt sites we showed neutrons dynamics (and not derived soil moisture) since the aim was to compare the performance of Finapp sensor to more conventional neutron sensor. But we agree that a zoom-in can help understand possible differences and the Figure 5 below will replace in the new version of the manuscript the one in the original manuscript to better highlight the short-scale differences.*

[Figure]

*Figure 5. Comparison of neutron measured by Finapp and CRS 2000 at Marchfeld site (upper panels) and between Finapp and Lab-C at Marquardt.*

**RC: The authors argue that Jungfraujoch data "completely fails" in correcting neutrons. However, a single evidence is only visible for one (the second) out of many precipitation events (Fig 7). And only in San Pietro, while in Legnaro the performance of Jungfraujoch looks ok, i.e. a small increase of soil moisture is still evident for the small precipitation event. Unfortunatelly, other sites were omitted. And since no measurement of actual soil moisture is presented, any improvemtent of the muon correction over NM correction remains just speculation. It is necessary here to show either additional soil moisture data, or muon vs Jungfraujoch data, or muon vs global muon database data, and to discuss any other influencing effects, such as high atmospheric temperature. Alternatively, rephrase the claims of this study to be more speculative and subject to future work.**

*AC: we agree that within the present study we cannot be conclusive with the use of muons. In contrast, we believe that the use of muons for CRNS soil moisture is at the early stage and only more data and studies can shed some light on the topic. The time series collected during this study are also relatively short for testing this hypothesis and we consider the assessment of the muon correction beyond the scope of the present study. For this reason, we will better clarify within the new version of the manuscript that the topic is a subject of future work. Still we agree that a direct comparison with incoming neutron fluxes from, e.g., Jungfraujoch is valuable also here. We will add in the new version of the manuscript the figure described above and extend the discussion accordingly (see also response to community comment).*

**RC: A better description and discussion of the gamma detection mechanism is missing beyond "0.3-3.0 MeV". How is the signal separated from unwanted gamma rays (e.g., from other energies, or from the detection products from the reactions in the adjacent detectors)? Is there**

**an energy response function? What type of reactions are visible in this range beyond K, Th, and U, and what could be their source or influencing factors other than water accumulation? How is the instrument different from typical gamma-ray sensors used by national authorities or by Strati et al? How does the weak correlation to soil moisture compare to literature values?**

*AC: the gamma detector of the Finapp3 probe installed at these experimental sites is a standard commercial organic scintillator (EJ-200, from Eljen Technology Inc.). Due to the low effective atomic number $Z_{eff}$, typical of organic materials, gamma rays interact with the scintillator mainly by Compton scattering providing the typical spectrum shape of the Compton continuum from zero to the Compton edges (see figure 6 below). Compton edge positions allow also for a quite good linear calibration of the energy response.*

[Figure]

*Figure 6 (a and b), typical spectrum plastic scint. from Boo, J., Hammig, M.D. & Jeong, M. Compact lightweight imager of both gamma rays and neutrons based on a pixelated stilbene scintillator coupled to a silicon photomultiplier array. Sci Rep 11, 3826 (2021). https://doi.org/10.1038/s41598-021-83530-4;right typical env. gamma spectrum from K. Ford et al. "Remote Predictive Mapping 2. Gamma-Ray Spectrometry: A Tool for Mapping Canada's North", Geoscience Canada, 35(3-4); https://journals.lib.unb.ca/index.php/gc/article/view/11270*

*The typical gamma environmental spectrum is shown here above (figure 6 right): basically no gammas are present above 3 MeV. What is seen above this energy range are signals with larger energy deposit due mainly to cosmic muons: being the typical energy deposit about 2 MeV/cm, the average energy deposit of muons in a 2 inches detector is around 10 MeV, well above the gamma region, even when a partial quenching is present, producing a well separated bump in the high energy part of the spectrum. Finally, some counts are present in between the muon bump and the 3 MeV gamma limit. They are peripheral interactions from muons that crossed only part of the active volume. The total yield of these signals is anyhow negligible with respect to the total gamma yield. The manuscript will be integrated with this information to better explain the gamma detection.*

**Minor concerns**

**RC: Multiple use of the word "good" to describe the performance of the sensor (L45, L69, L70, L73, L230, 253). In a scientific publication -- in contrast to advertising material -- it is necessary to more concretely measure the performance, provide quality statistics and an uncertainty assessment. Sometimes numbers were given in the figures, but mostly missing in the text.**

*AC: we adopted a plenary style for the text and left the performance metrices in the figures. But to be more precise, we will also add these values in the text. Thank you for the suggestions.*

**RC: Improve description of the particle detection mechanisms and modules used, see comments below. Remember that your audience is GI.**

*AC: a detailed description of the detection mechanism were explained in* (Cester et al., 2016; Stevanato et al., 2019). *However, we agree that for sake of clarity a short description should be integrated also in the present manuscript. Specifically: incoming neutrons (fast) hit the soil and slow down by interacting with hydrogen. A part of them (slow) escape from the soil and can travel up to hundreds of meters before being absorbed. CRNS probes catch these neutrons moving above the soil. In the Fínapp detector, the Li-6 embedded inside the detectors has a large cross section for neutron capture. When a Li-6 nucleus captures a neutron, a nuclear reaction occurs and the compound Li-7 brakes into an alpha particle (He-4) and a triton (H-3) with a large energy release of almost 5 MeV. This energy is converted into light (a flash of optical photons) by the ZnS(Ag) crystals, one of the most common scintillator materials used also in the past in the cathode ray tubes to make the light spot where the electron beam was hitting the screen. Finally, a photo-multiplier tube (PMT) converts the light flash into an electrical signal, acquired by the digital electronics.*

**RC: Improve language to be more scientific and more concrete (not just "special property", "easily detect", "considred reliable", etc), see comments below.**

*AC: we will rephrase to be more concrete. See specific comment below. We will go through the manuscript and check additional text that needs further improvements.*

**RC: The authors claim to describe the "current state of the art of data processing", while sometimes they are missing out relevant and newer literature (see specific comments below). If the authors do not want to update their processing procedures with newer (but admittedly less established) methods -- which would be ok -- at least the above statement should be removed or properly discussed why the provided approaches were used.**

*AC: Thanks. We agree and we will remove the statement.*

**RC: The above comment is particularly relevant as the soil moisture presented in Fig 5 sometimes goes below zero, which is a common artefact of the Desilets et al 2010 approach, for instance.**

*AC: Thanks for the comments. We will add that in the discussion and refer to the newer literature for possible solutions.*

**RC: The authors seem to advertise a new excel tool to support CRNS calibration and data processing (Baroni et al 2022). Has it been peer reviewed elsewhere? I would reject to publish it as a supplement to this work, since it is very general and out of scope here, and explanation of the methods used were not given in the manuscript.**

*AC: we consider good practice to share the tools that have been used to process the data, when possible. The tools, in these cases, are two spreadsheets that have been developed to process soil*

*samples and neutron data. The tools integrated the methods cited in the manuscript and proposed by other authors* (Schrön et al., 2017). *For these reasons, we do not claim that these are new tools but Readers are welcome to check and use them. We also acknowledge that these tools could be considered by Readers as a starting base for CRNS data processing understanding but we also already cited in the first version of the manuscript more advanced tools in case Readers have bit more computer skills* (Power et al., 2021). *We will rephrase the statement in the manuscript to avoid misunderstanding.*

**RC: The authors argue that a dedicated soil sampling campaign should be conducted to the evaluate gamma-ray data. However, the soil sample campaigns performed for neutron calibration were partly taken in <25 m distance, I suppose from looking at Fig 3. Wouldn't this be helpful to evaluate the gamma footprint?**

*AC: as far as we are aware, the footprint for the gammas is smaller (25 m radius max considering the height of the installed sensor). Moreover, while it is expected that the sensitivity of signal decreases exponentially (horizontally and vertically), no analytic functions have been proposed and tested in literature to weights point-scale soil moisture measurements. For these reasons, we did not consider the soil samples as good references for an assessment of the gammas. In contrast, we believe that soil samples at much smaller distances should be collected. This is currently under investigation based on dedicated experiments. Overall, we acknowledge that the use of total gamma counts for soil moisture estimation is not the focus of the present study. In the new version of the manuscript we will focus on the use of this signal for discriminating precipitation and irrigation and we will only highlight in the discussion additional applications that could be assessed in future studies, i.e., soil moisture.*

**RC: The authors mention that the use of gamma rays for soil moisture estimation would "require additional refinements". For sure, this would be capabilities of a spectrometer, since total gamma rays are not well correlated as this study and as other literature has shown. Are spectrometric capabilities possible in future improvements of this device?**

*AR: Yes, it is possible to substitute the current standard plastic detector with an inorganic scintillator like NaI(Tl) to get a spectrometric response at a fair price. More expansive detectors with better spectroscopic features (like LBC) could also be considered. It should be evaluated however if a dedicated sensor would be better instead of a composite sensor like Finapp type. No upgrade of the Finapp device has been currently developed.*

**RC: Fig 8 shows only a short period of gamma data, shorter than the presented neutron data. So, is the measurement of three particles not simultaneous during the full period? If yes: isn't this the main key or if no, why not showing everything?**

*AR: Gammas are measured simultaneously by the sensor. Some data, however, have been lost due to setting in the electronic board and remote sensing transmission issues. These issues were fixed at the beginning of the experiments, but it was not possible to recover the previous data-sets. For this reason, neutrons time series in figure 8 starts earlier than gamma's. This information will be added in the new version of the manuscript.*

**Specific comments**

**RC: L32: replace "correct" by "reliable" or "accurate".**

*AR: thanks, we will replace with reliable.*

**RC: L47, L51: the correlation is "negative" rather than "inverse".**

*AR: thanks, we will replace with negative.*

**RC: L45-46: the citations list appears random, at least try to briefly explain which author found what and to what degree of certainty (not simply "good performance"). Use citations to concretely support your statements.**

*AR: thanks, we will re-arrange the citation by land-use and hydrological conditions.*

**RC: L48: "favourable" compared to what?**

*AR: thanks, we will rephrase in:*

*"providing the base for monitoring"*

**RC: L47: try to cite literature that actually decribe the physical process of neutron-hydrogen interactions.**

*AR: thanks. Here we will refer to studies focusing on soil moisture, biomass and snow application.*

**RC: L51: replace "noise" by "nuisance"**

*AR: thanks, we will replace.*

**RC: L54: rephrase or reorder: "1979" is not "decades later" than 1966, 2001, and 2021.**

*AR: we will change decades to years. Thank you.*

**RC: L60: add Köhli et al 2015 for up-to-date literature on the footprint.**

*AR: we will replace with this reference. Thank you.*

**RC: L63: what us "water management and assessment" and how do those cited papers contribute to it? Add also Evans et al 2016 for a reference to Cosmos-UK with regards to your mentioned national networks.**

*AR: thanks, we will add the COSMO-UK reference that is indeed a good example of how CRNS network can be integrated for supporting water management and assessment.*

**RC: L84: Use concrete and scientific language: how are scintillators "safer" than He-3?**

*AR: thanks, the term safer was intended to be related to boron 3 fluoride proportional gas tube. But we agree that the sentence was not clear and can be improved with more concrete and scientific language. We will add information about the material of a scintillator (see specific comment above)*

*and about the size in comparison to other commercial CRNS probe, and its advantage considering safety considerations in comparison to gas tubes.*

**RC: L85: Use concrete and scientific language: how are scintillators "relatively compact" compared to He-3 tubes?**

*AC: see comment above.*

**RC: L87: Use concrete and scientific language: what is a "special property"?**

*AR: we use the term "special property" to refer to the different light emission between a classical inorganic scintillator, typically used as spectrometric gamma detector (e.g. NaI(Tl)) and a plastic scintillator with pulse shape discrimination capability. The first one has fixed pulse shape, only amplitude changes proportionally to the energy released in the active volume, while the latter show different shapes depending on the interacting particles (typically lighter particles (gamma/electrons/muons) produce shorter pulses with respect to heavier particles (protons/tritons/alphas/). A pulse shape analysis can be therefore used to identify the type of interacting particle in these particular plastic scintillators. But we agree that term can be misleading, and we will extend the description based on the text above.*

**RC: L104: "easily detect", ok it is easy, but how exactly is the detection mechanism?**

*AR: the scintillator (ZnS(Ag) in this case) converts deposited energy in light, collected and converted into an electrical signal by the PMT. The energy release in the thin layers of the scintillator (a few hundreds of microns) is strong for local interactions coming from the n-Li capture reaction products, while is small for crossing muons and basically negligible for gamma rays. Consequently, the signal from a neutron capture reaction is "easily" detected, providing a large electrical signal, well above the typical voltage threshold used to cut the noise. This information will be added in the new version of the manuscript. Thanks for pointing this out.*

**RC: L106: this would be a good place to reference Fig 1a.**

*AC: thank you. We will add the reference here.*

**RC: L116-118: rephrase, as the paragraph is very unspecific and it is not clear what value the citations have provided. Please try to elaborate more on it, or remove the sentence. At least add "among others" to indicate an incomplete list.**

*AC: We will remove the sentence to avoid misunderstanding. Thank you.*

**RC: L118: rephrase "current state of the art" (see minor concerns above), as incoming correction does not account for rigidity (L125, Hawdon et al 2014), humidity correction ignores higher-mode dependency on soil moisture (L126, Köhli et al 2021), constant N0 ignore biomass dependency (Baatz et al 2014), and the neutron-moisture relationship might become invalid under dry conditions (eq 5, Köhli et al 2021). Not all of the approaches must be used here, as some have not been established yet, but the present approach should not be called "current ..."**

*AC: We will remove the sentence to avoid misunderstanding. Thank you.*

**RC: L130: note that it is risky to take mean values as references since they would change with every change of measurement period and are not transferrable to other places and times. Consider using constant values, e.g. from Bogena et al 2022.**

*AC: thanks for the suggestion. For the purpose of the current study the use of mean or a specific value would not affect the results. But we agree that for, e.g., an operational soil moisture network the use of constant values should be adopted. We will add a comment and the reference here.*

**RC: L136: since you refer to data processing procedures used at various sites in Europe, it would be a good place to cite Bogena et al 2022.**

*AC: Thank you. We will add this reference here.*

**RC: L148, rephrase "has been shown to be an alternative", since Stevanato et al may have provided first evidence, but final proof is still missing. Your new sensor could certainly contribute to collecting more data to better investigate this hypothesis in the future.**

*AC: Thank you. We agree that the use of muons for CRNS correction is at the early stage and it is beyond the scope of the present study to derive any conclusions. But we appreciate the comment that this type of sensor can contribute to collecting more data to better investigate this hypothesis and we will integrate this discussion in the new version of the manuscript.*

**RC: L149: be more careful here, you are presenting a new measurement method, but you cannot claim that this methodological approach actually works. Please discuss potential effects of atmospheric temperature, and suggest further research as this is out of scope here.**

*AC: Thanks. We agreed and better highlight assumptions, possible improvements, and further tests. See also general comments above.*

**RC: L157: do you mean air temperature instead of air humidity?**

*AC. Yes, thank you. It will be corrected in the new version of the manuscript.*

**RC: L60: remove second appearance of the link.**

*AC. Thank you. It will be corrected in the new version of the manuscript.*

**RC: L171: "this energy region", be more specific please, which region exactly?**

*AR: the characteristic 1.46 MeV gamma ray from K-40 decay is seen in a plastic scintillator mainly by Compton scattering interactions and generates therefore a sort of plateau from 0 to about 1.2 MeV, with an increased yield at the high edge (Compton Edge). The gamma counts in the spectrum with energies between about 1.0 MeV to 2.5 MeV came mostly from Compton interactions of K-40, Bi-214 and Tl-208 present in the soil (and therefore are affected by water attenuation). We will add this information in the new version of the manuscript.*

**RC: L176: do you mean "total signal" instead of "these signals"?**

*AR: yes, thank you. It will be corrected.*

**RC: L198: in the discussion about meter-scale influences on neighboring sensors, add Schrön et al 2018 (GI) and Patrignani et al 2021 (Frontiers) who discussed exactly that.**

*AR: These references will be added. Thank you.*

**RC: Fig 2: add distances, meter scales, and more of the surrounding to better support your discussion on nearby influences.**

*AR: Fig 2 shows the installation of the sensors at Marchfeld and at Marquardt. We did not discuss the influences of the surrounding and it is not clear for us the comment. Maybe the Reviewer is pointing to figure 3? In that case the figure will be replaced, and more information will be integrated also based on additional comments that have been provided. See comment below.*

**RC: L195: do you mean "long-term obervations and real-time data transmission"?**

*AR: yes, thank you. We will correct.*

**RC: L233: please rephrase, relative to Lab-C?**

*AR: We will rephrase clarifying that neutron counting rate for Finapp3 is lower than both CRS2000 and Lab-C and thus it has higher statistical noise. Still when the signal is integrated over 6 hours and smoothed, the dynamic is well in agreement. Please note that the figure 4 will be also replaced with a zoom-in to show short-scale dynamics as suggested in the other comment.*

**RC: L235: "can be considered reliable", at this point of the text no mesures have been shown. There are measures in the Fig 4 legend, though, which should be mentioned in the text before claiming something reliable.**

*AR: yes, thanks. We will first indicate the measures and then comments.*

**RC: L253: Same as above, please mention and discuss RMSE before calling something reliable.**

*AR: yes, thanks. We will first indicate the measures and then comments.*

**RC: L287: consider adding Jakobi et al 2018.**

*AR: yes, thanks. We will add that.*

**RC: L300: the improvement seems to be only during the second event, not for all "precipitation events".**

*AR: The discussion of the muons will be completed revised with more details and data as discussed above. This statement will be removed and modified accordingly.*

**RC: Fig 8: what is the cause of the spike of the correlation plot? Can you add measures such as R squared?**

*AR: the spike is the wash out effect: the peak in the total gamma radiation generated by the deposition of atmospheric radon during the precipitation events. Please note that based on the overall comments provided also from the other Reviewer, we have decided to focus more on the use of total gamma for showing the capability to discriminate precipitation and irrigation event. In contrast, the use of total gammas for soil moisture estimation will be only mentioned and considered for further studies.*

**RC: L331: put concrete measures and uncertainties rather than just "performed well".**

*AC: we will add the measures also in the conclusion. Thank you for the comment.*

**Cosmetics**

**RC: Fig 3: use site names as panel titles instead of in a single legend. Color-blind people cannot identify which panel corresponds to what site.**

*AC: thank you. We will change the figure as below.*

[Figure]

**RC: Fig 4: rearrange or relabel, such that a,b are top and c,d are bottom (which is convention). Also, please put a comprehensive ylabel for c,d. In c,d, the two sensors are nit distinguishable for color-blind people, black/white displays or prints, etc. Use HCL color space and colors with different luminosity.**

*AC: we will change the figure as shown in the general comment above. Thank you for the suggestion.*

**RC:  Fig 5: use crosses instead of circles to better indicate the exact date and value of the data point. Also consider printing errorbars.**

*AC: we will change point shape with crosses instead of circles as in the figure below. The errorbars in the gravimetric soil samples are not added according to a discussion that emerged during the review processes for other papers. Specifically, errorbars can be calculated based on the soil samples. But the values are misinterpreted because Readers tend to consider that as errors instead of showing the spatial variability from the point-scale measurements. The figure should be read as average soil moisture over a certain footprint and the point (cross) represents exactly the average value. In contrast we do not have an estimation of the error of the average soil moisture and for this reason we prefer to not add an error bar.*

[Figure]

**References**

Cester, D., Lunardon, M., Moretto, S., Nebbia, G., Pino, F., Sajo-Bohus, L., Stevanato, L., Bonesso, I., Turato, F., 2016. A novel detector assembly for detecting thermal neutrons, fast neutrons and gamma rays. Nuclear Instruments and Methods in Physics Research Section A: Accelerators, Spectrometers, Detectors and Associated Equipment 830, 191–196. https://doi.org/10.1016/j.nima.2016.05.079

de Mendonça, R.R.S., Braga, C.R., Echer, E., Dal Lago, A., Munakata, K., Kuwabara, T., Kozai, M., Kato, C., Rockenbach, M., Schuch, N.J., Al Jassar, H.K., Sharma, M.M., Tokumaru, M., Duldig, M.L., Humble, J.E., Evenson, P., Sabbah, I., 2016. The temperature effect in secondary cosmic rays (muons) observed at the ground: analysis of the global muon detector network data. The Astrophysical Journal 830, 88. https://doi.org/10.3847/0004-637X/830/2/88

Dorman, L.I., 2004. Cosmic Rays in the Earth's Atmosphere and Underground, Astrophysics and Space Science Library. Springer Netherlands, Dordrecht. https://doi.org/10.1007/978-1-4020-2113-8

Maghrabi, A., Aldosary, A.F., 2018. The Effect of Some Meteorological Parameters on the Cosmic Ray Muons detected by KACST detector, in: Proceedings of 35th International Cosmic Ray Conference — PoS(ICRC2017). Presented at the 35th International Cosmic Ray Conference, SISSA Medialab, p. 062. https://doi.org/10.22323/1.301.0062

Power, D., Rico-Ramirez, M.A., Desilets, S., Desilets, D., Rosolem, R., 2021. Cosmic-Ray neutron Sensor PYthon tool (crspy 1.2.1): an open-source tool for the processing of cosmic-ray neutron and soil moisture data. Geoscientific Model Development 14, 7287–7307. https://doi.org/10.5194/gmd-14-7287-2021

Schrön, M., Köhli, M., Scheiffele, L., Iwema, J., Bogena, H.R., Lv, L., Martini, E., Baroni, G., Rosolem, R., Weimar, J., Mai, J., Cuntz, M., Rebmann, C., Oswald, S.E., Dietrich, P., Schmidt, U., Zacharias, S., 2017. Improving calibration and validation of cosmic-ray neutron sensors in the light of spatial sensitivity. Hydrol. Earth Syst. Sci. 21, 5009–5030. https://doi.org/10.5194/hess-21-5009-2017

Stevanato, L., Baroni, G., Cohen, Y., Cristiano Lino, F., Gatto, S., Lunardon, M., Marinello, F., Moretto, S., Morselli, L., 2019. A Novel Cosmic-Ray Neutron Sensor for Soil Moisture Estimation over Large Areas. Agriculture 9, 202. https://doi.org/10.3390/agriculture9090202

Stevanato, L., Baroni, G., Oswald, S.E., Lunardon, M., Mares, V., Marinello, F., Moretto, S., Polo, M., Sartori, P., Schattan, P., Ruehm, W., 2022. An Alternative Incoming Correction for Cosmic-Ray Neutron Sensing Observations Using Local Muon Measurement. Geophysical Research Letters 49, e2021GL095383. https://doi.org/10.1029/2021GL095383

---

## Author Response (AR1)

Testing a novel sensor design to jointly measure cosmic-ray neutrons, muons and gamma rays for non-invasive soil moisture estimation by Gianessi et al. https://doi.org/10.5194/gi-2022-20

**Author Response to Reviewer #1**

**RC:** *Reviewer Comment*, *AR: Author Response,* ⬜ Manuscript text

**RC: The authors present a novel set of experiments using a new sensor to simultaneously measure neutrons, muons, and gammas. The new sensor is compared to conventional sensors with satisfactory results for longer time periods of integration (i.e. 6 hrs for neutrons). The technology is lighter and potentially a lower cost, which will open up doors for more applications in science and in practical applications. The manuscript is well written and appropriate for the journal. I have a few suggestions that should be addressed prior to publication. Some instances of English grammar and word choice will need to be addressed.**

*AR: Thank you for the very quick and very positive feedback. All the comments and suggestions have been implemented in the new version of the manuscript as described in the point-by-point response below.*

**RC: L33: "Runoff generation"**

*AR: Thanks, corrected.*

**RC: L 40: "More recently, attention"**

*AR: Thanks, corrected.*

**RC: L 44: "has shown"**

*AR: Thanks, corrected.*

**RC: Figure 1: Are events outside of the blue and red ovals not included in the analysis?**

*AR: Figure 1 is only descriptive. The PSD vs. integrated charge plane is the simplest way to show where most of the neutrons and muons lie. In contrast, the actual selection of the events is based on the analysis of statistical outliers over different parameters. The text in the new version of the manuscript has been modified as follows to better clarify this aspect. Please also note that Figure 1 has been replaced to improve the readability.*

> Among others, a typical parameter used in this analysis is the so-called pulse-shape-discrimination parameter (PSD), given by the ratio of the integrated charge in the tail of the signal with respect to the total integrated charge. An example is shown in Figure 1a, which shows how different particles (here thermal neutrons and cosmic muons) populate very different regions in the PSD *vs.* integrated-charge plane. For more details on the analysis and on the parameters used for the identification of the single events we refer to more specific studies (e.g., Cester et al., 2016).

[Figure]

Figure 1. (a) Typical Pulse Shape Discrimination (PSD) *vs.* integrated charge plot for a FINAPP3 detector. Red and blue ovals indicate the neutron and muon region respectively; (b) scintillator-based sensor FINAPP3 with the two main detectors, photomultiplier (PMTs), board, and battery.

**RC: L165: For the gammas are there any corrections needed for pressure or air temperature/humidity variations?**

*AR: Based on current literature* (Baldoncini et al., 2018; Serafini et al., 2021; Strati et al., 2018)*, gammas do not require corrections for air pressure, air temperature and air humidity. We agree that this has some advantages in comparison to neutrons and we have added this information in the new version of the manuscript. Please note that, based on comments from Reviewer #2, all the discussion about gamma is now placed in a new section "2.5 Assessment of total gamma rays".*

> Noteworthy, the gamma signal should not be corrected for other effects (i.e., air pressure, air temperature and air humidity). For these reasons, it can provide some advantages to the use of neutrons for soil moisture application.

**RC: L247: I would use the SG filter on the neutron/gamma count time series, not the soil moisture time series that have been transformed by the calibration function.**

*AR: Thank you for the suggestion. We did not see relevant differences when applying the filter to the neutron/gamma time series or to the estimated soil moisture. But we acknowledge that the filter could be applied directly to the raw data. We have integrated this information into the new version of the manuscript as follows.*

> Before the transformation, the corrected hourly neutron values were smoothed with a Savitzky-Golay filter to decrease the random fluctuations at short time period as suggested in literature (Franz et al., 2020).

**RC: L 302-305: This sentence is confusing and long. Please rewrite.**

*AR: We have received from the Reviewer #2 and from Daniel Rasche (community comment) several comments and suggestions to improve the discussion about the use of muons for CRNS soil moisture estimation. Based on that, we have added in the new version of the manuscript many more details on the detected muon behavior and a comparison to the dynamic measured at a neutron monitoring data base. Accordingly, section 3.3 has been completely revised. We refer to this session of the new version of the manuscript to see the changes that have been integrated.*

**RC: Figure 7. Will be interesting to compare the Muon detection with the correction factor being proposed by McJannet and Desilets using cutoff rigidity and atmospheric depth with the NMDB historical data. Unfortunately, that work is in the review process still.**

*AR: Thank you for sharing this ongoing study. We have searched for this publication, and we have seen that the paper has just now been accepted. Indeed, a comparison between this new proposed correction approach and the use of muons would be very interesting. As discussed also in the response to Reviewer #2, our current time series, however, are relatively short and do not capture strong incoming variability. For this reason, we consider this comparison beyond the scope of the present study. In contrast, we have stressed in different parts of the new version of the manuscript that this new sensor design can provide, in a longer term, relevant data that can be used for testing different incoming correction approaches. The reference to the paper of McJannet and Desilets is also integrated in the conclusion. We refer to the new version of the manuscript to see the changes that have been integrated.*

**RC: Figure 8. So the soil moisture data is from FINAPP and not depth weighted following the Schron method? If you did have gravimetric surveys how would you depth weight them for the gamma sensitivity? From my understanding they would have a similar sensitivity with depth as the neutrons but be a little shallower (10-15cm?)?**

AR: *Yes, the Reviewer is right: soil moisture data is from the neutrons from Finapp. This estimated soil moisture is compared to total gammas. We are not aware of analytic functions that have been published to weight point-scale soil moisture data to correspond to gamma signal. However, current literature suggests exponential decreases in both vertical and horizonal direction* (Baldoncini et al., 2018). *So, we agree that gamma should have a similar sensitivity to neutrons but shallower depth and smaller footprint. We have added this information into the new manuscript to clarify how to better compare gamma signals to soil moisture series in future studies.*

> Among others, the weak correlation can be attributed to the smaller horizonal and vertical footprint of the gamma fluxes (<25 m radius, <15 cm depth) in comparison to the neutron (~100 m radius, ~40 cm depth). Thus, a dedicated soil sampling campaign within the theoretical soil volume detected by the gamma particles should be performed for better assessment. An exponential decrease of the sensitivity of the signal has also been suggested in literature in both horizontal and vertical directions (Baldoncini et al., 2018). However, considering that the gamma footprint is strongly affected by the height of the detector installation (van der Veeke et al., 2021), further and more dedicated experiments should be performed to develop specific weighting functions and to conduct a proper assessment.

**RC: Table 1. Iwema et al. 2015 recommends 3 calibrations for estimating N0. From the variability here I would do at least 3 to estimate some uncertainty on N0. I agree additional gravimetric studies are needed, particularly for establishing the gamma to soil moisture dependence, especially when used in cropping systems with significant temporal variations in vegetation biomass. Unfortunately, for CRNS and GRS studies all roads don't lead to Rome but to more gravimetric sampling :)**

**Iwema, J., Rosolem, R., Baatz, R., Wagener, T., & Bogena, H. (2015). Investigating temporal field sampling strategies for site-specific calibration of three soil moisture–neutron intensity parameterisation methods. HESS, 19, 3203–3216. https://doi.org/10.5194/hess-19-3203-2015**

AR: *Thank you for the comment and for the reference. We agree that soil surveys still play an important role in assessing calibration functions and validating modelling results. We have integrated this comment and this reference in the new version of the manuscript in section 3.2.*

> Anyway, these results confirm the need to conduct when possible more than one calibration campaign to account for some of these effects as it has been suggested in literature (Heidbüchel et al., 2016; Iwema et al., 2015).

**References**

Baldoncini, M., Albéri, M., Bottardi, C., Chiarelli, E., Raptis, K.G.C., Strati, V., Mantovani, F., 2018. Investigating the potentialities of Monte Carlo simulation for assessing soil water content via proximal gamma-ray spectroscopy. Journal of Environmental Radioactivity 192, 105–116. https://doi.org/10.1016/j.jenvrad.2018.06.001

Cester, D., Lunardon, M., Moretto, S., Nebbia, G., Pino, F., Sajo-Bohus, L., Stevanato, L., Bonesso, I., Turato, F., 2016. A novel detector assembly for detecting thermal neutrons, fast neutrons and gamma rays. Nuclear Instruments and Methods in Physics Research Section A: Accelerators, Spectrometers, Detectors and Associated Equipment 830, 191–196. https://doi.org/10.1016/j.nima.2016.05.079

Franz, T.E., Wahbi, A., Zhang, J., Vreugdenhil, M., Heng, L., Dercon, G., Strauss, P., Brocca, L., Wagner, W., 2020. Practical Data Products From Cosmic-Ray Neutron Sensing for Hydrological Applications. Front. Water 2. https://doi.org/10.3389/frwa.2020.00009

Heidbüchel, I., Güntner, A., Blume, T., 2016. Use of cosmic-ray neutron sensors for soil moisture monitoring in forests. Hydrology and Earth System Sciences 20, 1269–1288. https://doi.org/10.5194/hess-20-1269-2016

Iwema, J., Rosolem, R., Baatz, R., Wagener, T., Bogena, H.R., 2015. Investigating temporal field sampling strategies for site-specific calibration of three soil moisture–neutron intensity parameterisation methods. Hydrology and Earth System Sciences 19, 3203–3216. https://doi.org/10.5194/hess-19-3203-2015

Serafini, A., Albéri, M., Amoretti, M., Anconelli, S., Bucchi, E., Caselli, S., Chiarelli, E., Cicala, L., Colonna, T., De Cesare, M., Gentile, S., Guastaldi, E., Letterio, T., Maino, A., Mantovani, F., Montuschi, M., Penzotti, G., Raptis, K.G.C., Semenza, F., Solimando, D., Strati, V., 2021. Proximal Gamma-Ray Spectroscopy: An Effective Tool to Discern Rain from Irrigation. Remote Sensing 13, 4103. https://doi.org/10.3390/rs13204103

Strati, V., Albéri, M., Anconelli, S., Baldoncini, M., Bittelli, M., Bottardi, C., Chiarelli, E., Fabbri, B., Guidi, V., Raptis, K.G.C., Solimando, D., Tomei, F., Villani, G., Mantovani, F., 2018.

Modelling Soil Water Content in a Tomato Field: Proximal Gamma Ray Spectroscopy and Soil–Crop System Models. Agriculture 8, 60. https://doi.org/10.3390/agriculture8040060

van der Veeke, S., Limburg, J., Koomans, R.L., Söderström, M., de Waal, S.N., van der Graaf, E.R., 2021. Footprint and height corrections for UAV-borne gamma-ray spectrometry studies. Journal of Environmental Radioactivity 231, 106545. https://doi.org/10.1016/j.jenvrad.2021.106545

Testing a novel sensor design to jointly measure cosmic-ray neutrons, muons and gamma rays for non-invasive soil moisture estimation by Gianessi et al. https://doi.org/10.5194/gi-2022-20

**Author Response to Reviewer #2**

**RC:** *Reviewer Comment*, *AR: Author Response,* ☐ Manuscript text

**RC: The study presents a new sensor system to monitor epithermal neutrons, cosmic-ray muons, and total (i.e. non-spectrometric) gamma rays all in one device. The novelty of this work is not only the individual technology of the three components, it is rather the combination of them and its potential for research and applications. Hence, it is important to demonstrate that each of the three signals can be reliably and simultaneously measured, and how they can be used for the greater good.**

**The neutron measurements seem to correlate well with traditional neutron detector signals, at least the scale of several days to months. Although the idea of correcting incoming neutron radiation with local muon measurements is not yet established and still an active field of research, the presented sensor may provide a fantastic opportunity to collect a large data set that could help to falsify this hypothesis in the future. The rather high RMSE of soil moisture calibration data and neutron products (up to 10 m3/m3 , Fig 5) is not a big issue, since every neutron probe would probably see similar deviation considering the substatial spatial heterogeneity, biomass, etc. I highly appreciate the elaborate discussion of influencing factors led by the authors.**

**A major weakness of the study is that no muon data was shown and comparisons to existing muon and gamma data sets are missing. This hinders proper falsification of these two measurement approaches. Many claims made in this study require better rigorous support. If possible, I'd suggest to add data to the study that provides evidence of the proper functioning of all three detectors, since it is key to covince the readers about the reliable measurement of three different particles at once.**

**The work is highly relevant to the journal and will have a great impact in the science community. However, I suggest major revisions to better streamline the focus of the paper on falsification and evaluation of the three components, of which muons and gammas are yet insufficiently addressed. The detailed comments below could help to quickly address the missing pieces before publication.**

*AC: We thank the Reviewer for the overall positive feedback. On the one hand, the Reviewer highlighted some weaknesses in the study. On the other hand, he/she also gave us very clear details on how these limitations could be addressed. Specifically, we thank the Reviewer who appreciated the comparison and discussion about neutrons, and we agree that, in comparison, the assessment of gammas and muons have been less addressed. As discussed in more detail in the point-by-point response below, in the new version of the manuscript we have improved the assessment of the gammas by showing a comparison between data collected by the FINAPP3 sensor and by a gamma-ray spectrometer. Concerning the muons, we have shown a comparison with the neutron fluxes measured at a neutron monitoring station (e.g., Jungfraujoch), as also suggested in the Community Comment by Daniel Rasche (for details please also see Authors response to his comments). We have also extended the description and the discussion on the use of muons for soil moisture correction. Finally, we have better highlighted that the aim of the present study was not to fully address the use of muons for CRNS correction, but to show that this sensor can provide valuable data to test this hypothesis in future studies. We think that the manuscript has been strongly improved based on all these suggestions and changes and we are looking forward to further feedback.*

**Major concerns**

**RC: The authors present a new device which combines three existing measurement principles, but only evaluate one of them with conventional devices (here, only neutron counter vs. other CRNS probes). Since the main novelty is the availability of three detectors at the same time, I would have expected also a validation of (or at least evidence for) the proper functioning of the muon and gamma detectors. This could be easily done with existing gamma ray probes from national authorities, and muon telescopes or the global muon network. Also consider plotting measured muon time series together with Jungfraujoch data to identify differences in their response to cosmic rays.**

*AC: We agree with the Reviewer, and we thank you for the suggestion to add additional data and analyses for improving the assessment of muons and gammas. For the muon signal, we have integrated the discussion on the muon behavior and its dependences on air pressure and air temperature. Moreover, a comparison with incoming neutron fluxes has been presented and the discussion about the effect on the soil moisture estimation has been improved. For the gammas, a comparison with data collected by a standard gamma-ray spectrometer ([https://medusa-online.com/en/](https://medusa-online.com/en/)) installed at the Ceregnano site has been integrated. The potentiality of this signal for agro-environmental applications has been better visualized and discussed. The new version of the manuscript has been improved and extended accordingly in several parts. We refer to the new version of the manuscript for details and we only summarize below the main changes:*

- *In the description of the detector assembly (section 2.1), the capability to discriminate muon and gamma has been better explained and additional references have been cited.*
- *Two new sections have been added: "2.4 Assessment of muons counting rate" and "2.5 Assessment of total gamma rays" with the description on how these signals have been assessed.*
- *In the result section "3.3 On the use of muons for incoming corrections", Figure 7 has been replaced with two new Figures (Figure 7 and 8 below). The text has been extended with the description of the analysis and of the results (see section 3.3). Specifically, to better show the proper functioning of the muon detector, Figure 7 below shows the correlation between the relative muon vs. atmospheric pressure and air temperature at Legnaro site, as example. Figure 8 shows the comparison with incoming neutron fluxes and the effect on the estimated soil moisture.*

[Figure]

Figure 7. Comparison of data collected at Legnaro site: (a) relative air pressure vs. muon counting rate; (b) air temperature vs. corrected pressure muon counting rate.

[Figure]

Figure 8. The plots in the top row show the average precipitation over the four Italian experimental sites. The plots at the middle row show the incoming correction based on neutron monitoring station (JUNG) and based on the average muon detected at the four experimental sites. Standard deviation is also shown in grey area. Bottom plots show estimated soil moisture using the two different approaches for the incoming correction of the signal: based on the standard approach of data from neutron data base (e.g., JUNG plotted as orange line) and using locally detected muons (blue line).

- *In the result section "3.4 Assessment of measured total gamma rays", Figure 8 has been replaced with the new Figures 9 and 10 below. The discussion has been extended accordingly. Specifically, Figure 9 shows the correlation between total gamma counts measured by FINAPP3 and by the gamma ray spectrometer Medusa at different integration time (1 hour and 6 hours, respectively). Figure 10 better visualizes the use of the gamma signal to discriminate precipitation and irrigation events.*

[Figure]

Figure 9. Comparison between total gamma counts measured by FINAPP3 and Medusa gamma-ray spectrometer.

[Figure]

Figure 10. From top row, precipitation (blue) and irrigation (light blue) (mm d$^{-1}$), volumetric soil moisture estimated by FINAPP3 (m$^3$ m$^{-3}$) and total gamma counts (TGC) over the mean of the monitored period.

**RC: Muon processing is a bit vague.**

*AC: we have improved the description of the muon processing. We refer to the new version of the manuscript for details and we go through the Reviewer's comments to highlight the main changes.*

**Why is there no correction for atmospheric humidity on the muon signal?**

*AC: as reported in literature* (de Mendonça et al., 2016)*, muons are affected by pressure and air temperature. The effect of relative humidity has been investigated less, due to the absence of a relationship between the two variables as suggested by some researchers* (Dorman, 2004; Maghrabi and Aldosary, 2018)*. However, we cannot exclude this effect and we will also add this statement in the new version of the manuscript (see section 2.4).*

**Where do you get the temperature data from? Local near-surface temperature is probably not suited to represent temperature effects in the upper atmosphere.**

*The temperature has been measured at 2 m height. We agree that temperature measured at the high atmosphere (or the whole air temperature profile) should better account for this effect. Some studies found however that also local near surface temperature can be a good proxy* (de Mendonça et al., 2016)*. We are currently using this approach, but we acknowledge however that further comparisons are necessary to provide better quantification of the related uncertainty. This information has been added in the new version of the manuscript (see section 2.4).*

**The authors also mention that long-term local time series have been used to estimate the parameters, but the data is not shown and as they cite Stevanato et al, it is not clear whether this is valid for both, the German and the Italian sites.**

*AC: we extended the assessment of the muon signal by showing relation with air pressure and air temperature. See Figure 7 (above and in the new version of the manuscript). The muon-pressure relation agrees with previous study* (Stevanato et al., 2022)*. In contrast, the pressure-corrected muons relation does not identify a clear effect. This can be explained by the relatively small temperature range measured over the period (± 5°). For this reason, in the present study we use the parameters for pressure and temperature corrections that has been derived and presented by* (Stevanato et al., 2022)*. In that study, the parameters have been extracted by the analysis of the one-year muon data series taken in Padova (Italy), from November 2019 to October 2020. The sensor location is within a distance of around 200 km from our experimental sites and at similar altitude. For these reasons, we assumed in the present study that the parameters are valid. In contrast, we did not use muons correction for Germany and Austria sites where we agree that differences could be relevant and locally adjusted parameters would be preferrable. This discussion is added in the new version of the manuscript. See section 3.3.*

**RC: The authors present excellent agreement of their neutron data with conventional neutron detectors during 6 months. However, it is the short-scale effects which are particularly interesting, e.g. the coincident response to the onset of rain events, to dew formation, to atmospheric variations, to drying periods, etc. They could be different if detection energy or geometry differs, and could eventually shed light on the actual sensor performance. This is not identifiable from the presented data, at least not with the long x-scale chosen in Fig 4. I'd appreciate if the authors could provide a zoom-in of the data that allows for day-to-day analysis and comparison.**

*AC: Thank you for the suggestion. We have added in the new version of the manuscript the new Figure 4 below with a zoom-in of the data during a fast drop of the neutron signal to better highlight the short-*

*scale differences. Please only note that we have shown corrected neutron data and not derived soil moisture dynamic because the aim is to compare the performance of detecting neutron by the different detector. The discussion of the results has been extended accordingly (see section 3.1).*

[Figure]

Figure 4. Comparison of measured neutrons at Marchfeld site, Vienna, Austria (top row) and Marquardt site, Potsdam, Germany (bottom row) by the two different sensor pairs (CRS2000 and FINAPP3; Lab-C and FINAPP3). Plots (a) and (d) show the hourly values in orange and based on a running average of 6 hours (blue). Plots (b) and (e) show the neutron fluxes corrected for air pressure and with a running average of 6 hours. The relative counts over the mean are shown for comparison. Plot (c) and plot (f) show a zoom-in during a fast drop of the neutron counts.

**RC: The authors argue that Jungfraujoch data "completely fails" in correcting neutrons. However, a single evidence is only visible for one (the second) out of many precipitation events (Fig 7). And only in San Pietro, while in Legnaro the performance of Jungfraujoch looks ok, i.e. a small increase of soil moisture is still evident for the small precipitation event. Unfortunatelly, other sites were omitted. And since no measurement of actual soil moisture is presented, any improvemtent of the muon correction over NM correction remains just speculation. It is necessary here to show either additional soil moisture data, or muon vs Jungfraujoch data, or muon vs global muon database data, and to discuss any other influencing effects, such as high atmospheric temperature. Alternatively, rephrase the claims of this study to be more speculative and subject to future work.**

*AC: we agree with the Reviewer, and we have removed that sentence and reshaped the discussion. Specifically, we better emphasized in the new version of the manuscript that we cannot be conclusive with the use of muons within the present study. In contrast, we agree that the use of muons for CRNS soil moisture is at the early stage and only more data and studies can shed some light on the topic. The time series collected during this study are also relatively short for testing this hypothesis and we consider the assessment of the muon correction beyond the scope of the present study. See changes in 3.3 and in the conclusions. Still we agree that a direct comparison with incoming neutron fluxes from, e.g., Jungfraujoch is valuable also in this study. Figure 7 has been replaced with two new figures (see new*

*Figures 7 and 8 in the manuscript and reported also above). The discussion has been extended accordingly. See section 3.3.*

**RC: A better description and discussion of the gamma detection mechanism is missing beyond "0.3-3.0 MeV". How is the signal separated from unwanted gamma rays (e.g., from other energies, or from the detection products from the reactions in the adjacent detectors)? Is there an energy response function? What type of reactions are visible in this range beyond K, Th, and U, and what could be their source or influencing factors other than water accumulation? How is the instrument different from typical gamma-ray sensors used by national authorities or by Strati et al? How does the weak correlation to soil moisture compare to literature values?**

*AC: Thank you for the suggestion on how to improve the description and the discussion of the gamma signal. Accordingly, we have modified the manuscript in several parts. Description of the gamma detection is now better described in section (2.1). However, we also highlight that the gamma detection is based on a commercial scintillator. Thus, we believe that it is beyond the scope of the present study to present much more details. Instead, some new references are added. We have also added a new section (2.5 Assessment of total gamma rays) where we describe how we assessed this signal. Section 3.4 has been revised accordingly with extended discussion.*

**Minor concerns**

**RC: Multiple use of the word "good" to describe the performance of the sensor (L45, L69, L70, L73, L230, 253). In a scientific publication -- in contrast to advertising material -- it is necessary to more concretely measure the performance, provide quality statistics and an uncertainty assessment. Sometimes numbers were given in the figures, but mostly missing in the text.**

*AC: thank you for the suggestion. We have modified the discussion first presenting some performance metrices followed by a more general discussion. Please see the changes in the results sections of the revised manuscript for details.*

**RC: Improve description of the particle detection mechanisms and modules used, see comments below. Remember that your audience is GI.**

*AC: The description has been extended. Please see the changes in the results sections of the revised manuscript for details.*

**RC: Improve language to be more scientific and more concrete (not just "special property", "easily detect", "considred reliable", etc), see comments below.**

*AC: we have rephrased several statements to be more concrete. We refer to the changes in the new version of the manuscript for details. Some specific changes are listed below.*

**RC: The authors claim to describe the "current state of the art of data processing", while sometimes they are missing out relevant and newer literature (see specific comments below). If the authors do not want to update their processing procedures with newer (but admittedly less established) methods -- which would be ok -- at least the above statement should be removed or properly discussed why the provided approaches were used.**

*AC: Thanks. We agree and we have removed the statement.*

**RC: The above comment is particularly relevant as the soil moisture presented in Fig 5 sometimes goes below zero, which is a common artefact of the Desilets et al 2010 approach, for instance.**

*AC: Thanks for the comments. The following statement has been added in the discussion of the soil moisture estimation (see section 3.2).*

> The use of more recently proposed soil moisture-neutron relation could also be tested to see possible compensation for these effects (Köhli et al., 2021).

**RC: The authors seem to advertise a new excel tool to support CRNS calibration and data processing (Baroni et al 2022). Has it been peer reviewed elsewhere? I would reject to publish it as a supplement to this work, since it is very general and out of scope here, and explanation of the methods used were not given in the manuscript.**

*AC: we consider good practice to share the tools that have been used to process the data, when possible. The tools, in these cases, are two spreadsheets that have been developed to process soil samples and neutron data. The tools integrated the methods cited in the manuscript and proposed by other Authors (Schrön et al., 2017). For these reasons, we do not claim that these are new tools but Readers are welcome to check and use them. We also acknowledge that these tools could be considered by Readers as a starting base for CRNS data processing but we also already cited in the first version of the manuscript more advanced tools in case Readers have bit more computer skills (Power et al., 2021). We have rephrased the statement in the manuscript to avoid misunderstanding as follows.*

> The data processing described above has been implemented in a simple spreadsheet available from (Baroni, 2022). For a more advanced data processing integrating also additional external data-sets readers can refer to (Power et al., 2021).

**RC: The authors argue that a dedicated soil sampling campaign should be conducted to the evaluate gamma-ray data. However, the soil sample campaigns performed for neutron calibration were partly taken in <25 m distance, I suppose from looking at Fig 3. Wouldn't this be helpful to evaluate the gamma footprint?**

*AC: yes, the Reviewer is right that we have collected samples at short distance that might be used for the assessment of the moisture-gamma relation. However, we still believe that this analysis would require some more insight on the signal response to environmental conditions and the data collected are limited or difficult to compare. Specifically, a soil moisture-gamma relation has been derived in literature but only for 40-K or Tl-208 peaks (Baldoncini et al., 2018). So, most likely the comparison of these relations with our measured total gamma counts is misleading. Moreover, the sensitivity of the signal has been found to decrease exponentially (horizontally and vertically) with a strong dependency to the height of the installation of the sensor (van der Veeke et al., 2021). No analytic functions have been proposed however as far as we are aware. For this reason, a direct comparison with point-scale soil moisture is not straightforward. Moreover, few field studies have been reported in literature about the use of gamma for soil moisture estimation. For this reason, we think that a greater number of samples over short distances and more campaigns should be performed to test this relation. Some modelling scenarios should also be performed when possible to provide some new insights and support the empirical results. For these reasons, we believe that this analysis is beyond the scope of the present study, and it has not been integrated in the new version of the manuscript.*

*Instead, in the new version of the manuscript, we have focused the discussion on the description of the data collected and we have added the quantitative comparison to a gamma ray spectrometer (Medusa).*

*The signal is further described in comparison to the environmental conditions (precipitation, irrigation and neutron derived soil moisture). The discussion has been extended accordingly. We refer to the changes in section 3.4 of the new version of the manuscript for details.*

**RC: The authors mention that the use of gamma rays for soil moisture estimation would "require additional refinements". For sure, this would be capabilities of a spectrometer, since total gamma rays are not well correlated as this study and as other literature has shown. Are spectrometric capabilities possible in future improvements of this device?**

*AR: Yes, it is possible to substitute the current standard plastic detector with an inorganic scintillator like NaI(Tl) to get a spectrometric response at a fair price. More expansive detectors with better spectroscopic features (like LBC) could also be considered. It should be evaluated however if a dedicated sensor would be better instead of a composite sensor like FINAPP3 type. No upgrade of the FINAPP device has been currently developed.*

**RC: Fig 8 shows only a short period of gamma data, shorter than the presented neutron data. So, is the measurement of three particles not simultaneous during the full period? If yes: isn't this the main key or if no, why not showing everything?**

*AR: Gammas are measured simultaneously by the sensor. Some data, however, have been lost due to setting in the electronic board and remote sensing transmission issues. These issues were fixed at the beginning of the experiments, but some data have been deprecated. For this reason, neutrons time series in figure 8 starts earlier than gamma's. Muons suffered the same problems. This information has been added in the new version of the manuscript. We refer to the new version of the manuscript for details.*

**Specific comments**

**RC: L32: replace "correct" by "reliable" or "accurate".**

*AR: thanks, we have replaced with "accurate".*

**RC: L47, L51: the correlation is "negative" rather than "inverse".**

*AR: thanks, we have replaced with negative.*

**RC: L45-46: the citations list appears random, at least try to briefly explain which author found what and to what degree of certainty (not simply "good performance"). Use citations to concretely support your statements.**

*AR: thanks, we have arranged the citations by land-use and hydrological conditions. We believe there is no need to summarize more details about the results in each study but Readers are pointed to the specific paper for details.*

**RC: L48: "favourable" compared to what?**

*AR: thanks, we have rephrased in: "providing the base for monitoring"*

**RC: L47: try to cite literature that actually decribe the physical process of neutron-hydrogen interactions.**

*AR: thanks. Here we rephrase organizing the studies with the focus on soil moisture, biomass and snow applications.*

**RC: L51: replace "noise" by "nuisance"**

*AR: thanks, corrected.*

**RC: L54: rephrase or reorder: "1979" is not "decades later" than 1966, 2001, and 2021.**

*AR: Thank you. Corrected with "some years later".*

**RC: L60: add Köhli et al 2015 for up-to-date literature on the footprint.**

*AR: Reference has been added. Thank you.*

**RC: L63: what us "water management and assessment" and how do those cited papers contribute to it? Add also Evans et al 2016 for a reference to Cosmos-UK with regards to your mentioned national networks.**

*AR: thanks, we have also added Evans et al. that is indeed a good example of how CRNS network can be integrated for supporting water management and assessment. We believe it is beyond the scope of the paper to explain in more details how soil moisture observations can contribute to water management and assessment.*

**RC: L84: Use concrete and scientific language: how are scintillators "safer" than He-3?**

*AR: thank you. The term "safer" was intended to be related to boron 3 fluoride proportional gas tube. But we agree that the sentence was not clear and has been rephrased as follows.*

> The main advantages are the use of cheaper and safer materials than proportional gas tubes based on helium-3 or boron trifluoride, respectively.

**RC: L85: Use concrete and scientific language: how are scintillators "relatively compact" compared to He-3 tubes?**

*AC: thank you. The sentence has been rephrased as follows.*

> Moreover, the flexibility in manipulating the detecting material (e.g., thin layers) allows to optimize the sensitive area and to develop relatively efficient but compact sensors.

**RC: L87: Use concrete and scientific language: what is a "special property"?**

*AC: thank you. The sentence has been rephrased as follows.*

> The scintillator materials used for neutron detection, in particular, have the unique property in comparison to inorganic scintillator to release the light in different ways when hit by different particles.

**RC: L104: "easily detect", ok it is easy, but how exactly is the detection mechanism?**

*AR: the description of the detection mechanism has been extensively revised. We refer to section 2.1 of the new version of the manuscript for details on the changes.*

**RC: L106: this would be a good place to reference Fig 1a.**

*AC: thank you. We have added the reference to the figure here.*

**RC: L116-118: rephrase, as the paragraph is very unspecific and it is not clear what value the citations have provided. Please try to elaborate more on it, or remove the sentence. At least add "among others" to indicate an incomplete list.**

*AC: We have removed the sentence to avoid misunderstanding. Thank you.*

**RC: L118: rephrase "current state of the art" (see minor concerns above), as incoming correction does not account for rigidity (L125, Hawdon et al 2014), humidity correction ignores higher-mode dependency on soil moisture (L126, Köhli et al 2021), constant N0 ignore biomass dependency (Baatz et al 2014), and the neutron-moisture relationship might become invalid under dry conditions (eq 5, Köhli et al 2021). Not all of the approaches must be used here, as some have not been established yet, but the present approach should not be called "current ..."**

*AC: We have removed the sentence to avoid misunderstanding. Thank you.*

**RC: L130: note that it is risky to take mean values as references since they would change with every change of measurement period and are not transferrable to other places and times. Consider using constant values, e.g. from Bogena et al 2022.**

*AC: thanks for the suggestion. For the purpose of the current study the use of mean or a specific value would not affect the results. But we agree that for, e.g., an operational soil moisture network the use of constant values should be adopted. We have changed as follows:*

> *href*, *pref* and *Iref* are reference values (here the average is taken) of air pressure, absolute air humidity and incoming neutron flux during the measuring period, respectively.

**RC: L136: since you refer to data processing procedures used at various sites in Europe, it would be a good place to cite Bogena et al 2022.**

*AC: Thank you. We have added this reference here.*

**RC: L148, rephrase "has been shown to be an alternative", since Stevanato et al may have provided first evidence, but final proof is still missing. Your new sensor could certainly contribute to collecting more data to better investigate this hypothesis in the future.**

*AC: Thank you. We agree that the use of muons for CRNS correction is at the early stage and we are not conclusive with the data collected within this study. But we appreciate the comment that this type of sensor can contribute to collect more data to better investigate this hypothesis. The sentence has been rephrased as follows and we have integrated this discussion in the new version of the manuscript. Please also see conclusion section.*

> The use of muons has been shown to be a possible alternative to the use of the neutron monitoring stations for incoming correction since they are produced from the same cascade as

> cosmic-ray induced neutrons in the atmosphere (Stevanato et al., 2022).

**RC: L149: be more careful here, you are presenting a new measurement method, but you cannot claim that this methodological approach actually works. Please discuss potential effects of atmospheric temperature, and suggest further research as this is out of scope here.**

*AC: Thanks. We agreed and better highlight assumptions, possible improvements, and further tests. See also general comments above.*

**RC: L157: do you mean air temperature instead of air humidity?**

*AC. Thank you. Here we refer to air pressure and air temperature. The sentence has been corrected accordingly.*

**RC: L160: remove second appearance of the link.**

*AC. Thank you. Corrected.*

**RC: L171: "this energy region", be more specific please, which region exactly?**

*AR: the description has been improved as follows.*

> For this reason, the gamma-ray signal (i.e., the 40K full-energy peak at 1.46 MeV or, anyhow, in the energies between about 1.0 MeV to 2.5 MeV) shows a negative correlation with the amount of water in the soil and thus this relation can be used to estimate soil moisture dynamic (Strati et al., 2018).

**RC: L176: do you mean "total signal" instead of "these signals"?**

*AR: The sentence has been removed and a new section 2.4 Assessment of muon and total gamma rays" has been integrated with more details about the comparison. Please see the new version of the manuscript for details on the changes.*

**RC: L198: in the discussion about meter-scale influences on neighboring sensors, add Schrön et al 2018 (GI) and Patrignani et al 2021 (Frontiers) who discussed exactly that.**

*AR: These references have been added. Thank you.*

**RC: Fig 2: add distances, meter scales, and more of the surrounding to better support your discussion on nearby influences.**

*AR: Fig 2 shows the installation of the sensors at Marchfeld and at Marquardt. We did not discuss the influences of the surroundings for these experimental sites. In contrast, this has been discussed for the Italian sites. A new Figure 3 has been integrated in the new version of the manuscript with a better legend and scale. We hope that this will address the comment of the Reviewer.*

**RC: L195: do you mean "long-term obervations and real-time data transmission"?**

*AR: thank you. Corrected.*

**RC: L233: please rephrase, relative to Lab-C?**

*AR: We have rephrased as follow:*

> the relative lower sensitivity of FINAPP3 produced a higher amount of statistical noise when compared to its benchmark (CRS2000 or Lab-C, respectively).

**RC: L235: "can be considered reliable", at this point of the text no mesures have been shown. There are measures in the Fig 4 legend, though, which should be mentioned in the text before claiming something reliable.**

*AR: thanks. We have reorganized the section presenting first the performance measures followed by more general comments. We refer to the new version of the manuscript for details about the changes.*

**RC: L253: Same as above, please mention and discuss RMSE before calling something reliable.**

*AR: Thanks. Corrected as before.*

**RC: L287: consider adding Jakobi et al 2018.**

*AR: thanks. Reference has been added.*

**RC: L300: the improvement seems to be only during the second event, not for all "precipitation events".**

*AR: The discussion of the muons has been completely revised with more details and data as discussed above. This statement has been modified accordingly. Please see the new version of the manuscript for details.*

**RC: Fig 8: what is the cause of the spike of the correlation plot? Can you add measures such as R squared?**

*AR: the spike is the wash out effect: the peak in the total gamma radiation generated by the deposition of atmospheric radon during the precipitation events. We have calculated the Pearson correlation coefficient (r =-0.18) to highlight the negative correlation. This value has been added to the new version of the manuscript and the discussion extended as follows.*

> the total gamma counts show higher dynamic at sub-daily time scale in comparison to the estimated neutron-based soil moisture and the correlation between the signals is weak (Pearson correlation coefficient r = -0.18).

**RC: L331: put concrete measures and uncertainties rather than just "performed well".**

*AC: we have added the measures also in the conclusion. Thank you for the comment.*

**Cosmetics**

**RC: Fig 3: use site names as panel titles instead of in a single legend. Color-blind people cannot identify which panel corresponds to what site.**

*AC: thank you. We have changed the figure as below.*

[Figure]

**RC: Fig 4: rearrange or relabel, such that a,b are top and c,d are bottom (which is convention). Also, please put a comprehensive ylabel for c,d. In c,d, the two sensors are nit distinguishable for color-blind people, black/white displays or prints, etc. Use HCL color space and colors with different luminosity.**

*AC: we have changed the figure as shown in the general comment above. Thank you for the suggestion.*

**RC: Fig 5: use crosses instead of circles to better indicate the exact date and value of the data point. Also consider printing errorbars.**

*AC: we have changed point shape with crosses instead of circles. The errorbars in the gravimetric soil samples are not added according to a discussion that emerged during the review processes for other papers. Specifically, errorbars can be calculated based on the soil samples. But the values are misinterpreted because Readers tend to consider that as errors instead of showing the spatial variability from the point-scale measurements. The figure should be read as average soil moisture over a certain footprint and the point (cross) represents exactly the average value. In contrast we do not have an estimation of the error of the average soil moisture and for this reason we prefer to not add an error bar.*

[Figure]

**References**

Baldoncini, M., Albéri, M., Bottardi, C., Chiarelli, E., Raptis, K.G.C., Strati, V., Mantovani, F., 2018. Investigating the potentialities of Monte Carlo simulation for assessing soil water content via proximal gamma-ray spectroscopy. Journal of Environmental Radioactivity 192, 105–116. https://doi.org/10.1016/j.jenvrad.2018.06.001

Baroni, G., 2022. Spreadsheets for soil samples and CRNS data processing. https://doi.org/10.5281/zenodo.7156607

de Mendonça, R.R.S., Braga, C.R., Echer, E., Dal Lago, A., Munakata, K., Kuwabara, T., Kozai, M., Kato, C., Rockenbach, M., Schuch, N.J., Al Jassar, H.K., Sharma, M.M., Tokumaru, M., Duldig, M.L., Humble, J.E., Evenson, P., Sabbah, I., 2016. The temperature effect in secondary cosmic rays (muons) observed at the ground: analysis of the global muon detector network data. The Astrophysical Journal 830, 88. https://doi.org/10.3847/0004-637X/830/2/88

Dorman, L.I., 2004. Cosmic Rays in the Earth's Atmosphere and Underground, Astrophysics and Space Science Library. Springer Netherlands, Dordrecht. https://doi.org/10.1007/978-1-4020-2113-8

Maghrabi, A., Aldosary, A.F., 2018. The Effect of Some Meteorological Parameters on the Cosmic Ray Muons detected by KACST detector, in: Proceedings of 35th International Cosmic Ray Conference — PoS(ICRC2017). Presented at the 35th International Cosmic Ray Conference, SISSA Medialab, p. 062. https://doi.org/10.22323/1.301.0062

Power, D., Rico-Ramirez, M.A., Desilets, S., Desilets, D., Rosolem, R., 2021. Cosmic-Ray neutron Sensor PYthon tool (crspy 1.2.1): an open-source tool for the processing of cosmic-ray neutron and soil moisture data. Geoscientific Model Development 14, 7287–7307. https://doi.org/10.5194/gmd-14-7287-2021

Schrön, M., Köhli, M., Scheiffele, L., Iwema, J., Bogena, H.R., Lv, L., Martini, E., Baroni, G., Rosolem, R., Weimar, J., Mai, J., Cuntz, M., Rebmann, C., Oswald, S.E., Dietrich, P., Schmidt, U., Zacharias, S., 2017. Improving calibration and validation of cosmic-ray neutron sensors in the light of spatial sensitivity. Hydrol. Earth Syst. Sci. 21, 5009–5030. https://doi.org/10.5194/hess-21-5009-2017

Stevanato, L., Baroni, G., Oswald, S.E., Lunardon, M., Mares, V., Marinello, F., Moretto, S., Polo, M., Sartori, P., Schattan, P., Ruehm, W., 2022. An Alternative Incoming Correction for Cosmic-Ray Neutron Sensing Observations Using Local Muon Measurement. Geophysical Research Letters 49, e2021GL095383. https://doi.org/10.1029/2021GL095383

van der Veeke, S., Limburg, J., Koomans, R.L., Söderström, M., de Waal, S.N., van der Graaf, E.R., 2021. Footprint and height corrections for UAV-borne gamma-ray spectrometry studies. Journal of Environmental Radioactivity 231, 106545. https://doi.org/10.1016/j.jenvrad.2021.106545

Testing a novel sensor design to jointly measure cosmic-ray neutrons, muons and gamma rays for non-invasive soil moisture estimation by Gianessi et al. https://doi.org/10.5194/gi-2022-20

**Author Response to Community Comment #1**

CC: *Community Comment*, *AR: Author Response*

**CC1: Dear authors,**

**I think that combining observations of neutrons for soil moisture estimation, gamma rays for distinguishing rainfall and irrigation as well as muons for correcting neutron observations as suggested by Stevanato et al. (2022) in a single sensor system is very interesting and would be a great advantage from a science and application perspective.**

**Cosmic-ray neutrons of different energies (e.g. Hubert et al. 2019) as well as muons (e.g. Braun et al. 2009) respond to solar events. As your observation period covers such an event, it poses an excellent opportunity to evaluate the muon data and their use for the correction of epithermal neutron observations. Thus, I suggest to add a time series plot covering the solar event (+/- a few weeks) showing the neutron monitor time series of the closest neutron monitors Jungfraujoch (JUNG) and Athens (ATHN) and the muon data of the four observation sites. A similar response to the solar event would underline the suitability of the sensor's muon product and the suggested correction approach.**

**Kind regards,**

**Daniel Rasche**

*AR: Dear Daniel Rasche, thank you for the comments and for the references. Based on your suggestions and from the comments of Reviewer #2, we have analyzed in more detail the dynamic recorded by neutron monitors (NMDBs) and the muons during our experiments. Figure 7 has been replaced with two new Figures (see Figure 7 and 8 in the new version of the manuscript). The text has been extended with the description of the analysis and of the results (see section 3.3). Specifically, Figure 8 reported also below shows the comparison with incoming neutron fluxes and the effect on the estimated soil moisture, as suggested. We refer to the new version of the manuscript for the new discussion that has been integrated. However, we want to underline also here that it was not the aim of the present study to conclude about the use of muons for CRNS soil moisture correction. In contrast, as discussed also in literature, this approach is at the early stage and only additional data will support this hypothesis and further developments. As also suggested by the Reviewer #2, we show that sensors like the one presented in this study provides an excellent base to collect these new data. We have rephrased any statements that could have been misleading into the new version of the manuscript and better clarified the focus of the present study.*

[Figure]

Figure 8. The plots in the top row show the average precipitation over the four Italian experimental sites. The plots at the middle row show the incoming correction based on neutron monitoring station (JUNG) and based on the average muon detected at the four experimental sites. Standard deviation is also shown in grey area. Bottom plots show estimated soil moisture using the two different approaches for the incoming correction of the signal: based on the standard approach of data from neutron data base (e.g., JUNG plotted as orange line) and using locally detected muons (blue line).

---

## Author Response (AR2)

Testing a novel sensor design to jointly measure cosmic-ray neutrons, muons and gamma rays for non-invasive soil moisture estimation by Gianessi et al. https://doi.org/10.5194/gi-2022-20

**Author Response to Editor**

*Comments from the Editor:*

*Dear Authors, Both reviewers have appreciated the revised version of your contribution to GI journal. Few suggestions are to be addressed. Best Regards, Jean Dumoulin*

*Dear Authors, Thank you for your revised version and the quality of the research work you made. I encourage you to take into consideration the last suggestions of reviewers in your final version. There are a few open issues, which, however, could be solved by simply fine-tuning the formulations in the text or some figures. I thank you in advance for this last effort. Best Regards, Jean Dumoulin*

Dear Editor,

thank you for handling the manuscript and the overall evaluation. We are glad to know that you and both Reviewers appreciated our effort to improve the quality of the manuscript. We went through the new comments of the second Reviewer, and we agree that the new suggestions are also very valuable. Based on that, we provide a point-by-point response and a new version of the manuscript where we have rephrased some text and further improved some figures accordingly.

Please also note that the first Author Stefano Gianessi does not work anymore at the University of Bologna (Italy) but he is now an employee of FINAPP company. For this reason, we have made some changes in the affiliation to be consistent. Please check if it is in line with the requirements of the Journal.

On behalf of the Authors,

Gabriele Baroni

Testing a novel sensor design to jointly measure cosmic-ray neutrons, muons and gamma rays for non-invasive soil moisture estimation by Gianessi et al. https://doi.org/10.5194/gi-2022-20

**Author Response to Reviewer #2**

RC: *Reviewer Comment,* AR: *Author Response,* ☐ Manuscript text

RC: *I appreciate the substantial changes to the manuscript based on the reviewer comments. There are a few remaining issues that I find not yet acceptable, but I see the paper on a good path to be published soon with minor revisions.*

AR: *Thank you for appreciating the scientific quality of the manuscript and for the additional suggestions to further improve the presentation quality. Point-by-point response to the comments is reported below. Based on that, we also provide a new version of the manuscript.*

RC: *Major*

RC: *- In general, I appreciate the conservative discussion of the results in the revised manuscript, but some formulations are still inadequate. The muon and gamma data do not show convincing agreement to scientifically expected results (low correlation to Medusa, no sign of FD events). At most, these results could be called "a first indication" that there is something behind it, but they need to be investigated in further studies (as was discussed, thank you). Still, I have to be strict here and point out some remaining inconsistencies that need to be reformulated to avoid misunderstanding and misinterpretation. Please see below.*

AR: *We put efforts to avoid misunderstanding and misinterpretation in the discussion of the results and we are glad of the acknowledgement of the Reviewer. These further suggestions are welcome and have been considered for further improvements.*

RC: *- Nov 2021 is a clear sign of a Forbush Decrease (FD). This effect has global influence on the neutron counts everywhere and should be corrected for all CRNS data. However, your Figure 8 indicates that the measured muons do not recognize the FD event, and are not very well in sync with the other features from JUNG, too. Hence, to me this is a clear indication that the measured muons are not suited for CRNS correction. You mentioned the reason that "FINAPP3 muon detector has been optimized to follow relative long-term variability (weeks to months)". How was this "optimization" done? Can't you show the original raw hourly muons flux? If the muon data is smoothed over many days, I won't consider it of any use for incoming neutron correction. In my view, this should be communicated as a major flaw of the method.*

AR: *We understand the concerns of the Reviewer, but we partially agreed with his/her opinion. On the one hand, these comments have been considered to better explain and discuss our results. Thus, some text in the manuscript has been modified accordingly as reported also below.*

*In the abstract, at L27 of the new version of the manuscript with track changes:*

> The muons and the total gamma-rays simultaneously detected by the sensor show promising features to account for the incoming variability and for discriminating irrigation and precipitation events, respectively.

*At L364 of the new version of the manuscript with track changes:*

> At the current stage, the reasons of these differences have been not identified but only some hypotheses are formulated. First, the FINAPP3  muons count rate is relatively low and the recorded signal is smoothed over relative long-time period (days) to reduce the statistical errors. For this reason, short term dynamics cannot be captured. Second, the muon detector is also not directional (e.g., as a telescope looking upward) but it measures muon particles that are scattered in all the directions.

*On the other hand, we believe that we cannot conclude that our data show **"a clear indication that the measured muons are not suited for CRNS correction"**, as suggested by the Reviewer and we disagree with his statement "**If the muon data is smoothed over many days, I won't consider it of any use for incoming neutron correction**". It should be acknowledged in fact that incoming correction should not only compensate FD events or other fast temporal changes. In contrast, seasonal to yearly fluctuations of the incoming neutron fluxes due to, e.g., solar cycles, are also very predominant effects, especially for establishing long-term soil moisture monitoring. In addition, we agree that FD events have effects on neutron fluxes everywhere, but it should also be highlighted that the effects are of different degree depending on the locations. Noteworthy, a recent paper about incoming correction based on NMDB showed the use of a proportional factor to properly scale the signal measured at, e.g., Jung station, to locations with different characteristics (McJannet and Desilets, 2023). In the light also of this publication, it should be underlined that we should not blindly accept the fluctuation detected at, e.g., Jung station, as the correct one. Overall, we believe that the Reviewer is opening a very interesting discussion. We fully agreed on rephrasing some statements to avoid misunderstanding or any speculations, but we believe that further data and analyses are needed to have "clear indication" of the use of muons for incoming correction for both short and long term fluctuations. We believe that this has been fully acknowledged in several parts of the manuscript (see for instance in the abstract at L29) and for this reason no further comments have now been included.*

***RC - The authors have responded to the request for a comparison of the gamma sensor with a traditional gamma sensor by adding section 3.4. Here, the authors mentiond that the correlation is low (R²=0.077) "mainly due to the presence of some outliers". This is not sound science. Please remove the outliers before publishing those results, or restrict the analysis to periods without precipitation. Otherwise readers cannot draw any conclusion from such a comparison.***

*AR: We believe that a comparison with raw data is still meaningful as it shows to the Readers an assessment if no post-processing is performed. For this reason, the plot and the discussion of the low correlation is not removed ($R^2$ value is only rounded to 0.08 to be consistent with the second reported value). But we agree and remove, however, these extreme values measured during precipitation events before smoothing the time series. The new correlation ($R^2 = 0.32$) is calculated now to this filtered and smoothed data. Figure and text have been changed as follows.*

> The correlation between the two signals is low at 1 h time resolution ($R^2$ = 0.08), mainly due to the presence of extreme values observed during the precipitation events. The correlation increases ($R^2$ = 0.32) with a consistent detected dynamic (Figure 8b) when these

extreme values are removed, and the time series is smoothed over  6 h  time window.

[Figure]

*RC- In the same section, the authors provide a 6h average of the data, present a correlation of R²=0.29, and conclude "the dynamic is well captured". Again with a super-tiny plot with grey-in-greyish lines (Fig 9, right) and no-transparent super-thick point markers clumping above each other (left). It is impossible to visually identify any details, shifts, biases, lag times, etc. Please update your general presentation of the results (see cosmetics). Please also avoid calling these results "well capturing the dynamics". A rough tendency to follow the Medusa data is visible, but the low correlation and the significant deviations indicate that the data products represent different environmental effects and should be used with care. Please be more clear about this in the text.*

*AR: Thank you for the suggestion. We put some more effort for improving figure 9 and updated the general presentation of the results as suggested. See answer to the comment above.*

*RC - The authors now mention that "air temperature measured at 2 m hight provides a good approximation on the muon effect (de Mendonça et al., 2016)". From a quick browse of the cited study, I see strong complexity in this relationship, with seasonal temperatures in upper and lower athmospheric zones being completely out of phase in some regions of the Earth, while being similar in others. From this the authors should conclude that this is a serious issue that should be investigated in the future, instead of making the impression that taking near-ground temperature is a good and generally applicable approach.*

*AR: Our statement reported by the Reviewer seems misleading (to us as well) if extracted by the context. For this reason, we report below the whole text.*

> Noteworthy, the whole air temperature profile should be considered for the correction. This would better represent the atmospheric condition and it would better capture the effect on muons. Some studies, however, have shown how the use of air temperature measured at 2 m hight provides a good approximation on the muon effect (de Mendonça et al., 2016). This approach is used also in this study, but it should be further tested in future research.

*It should be appreciated by the Reviewer by reading again the whole text, that we fully agree that the approach proposed by de Mendonça et al. is not taken for granted but should investigated in the future. In the discussion of the results this has also been highlighted. At L343 of the new version of the manuscript with track changes we state:*

> However, the representativeness of air temperature measured at 2 m hight in comparison to the need of a whole air temperature profile is also questionable and it should be further investigated (de Mendonça et al., 2016).

*Overall, we stated that the approach proposed by de Mendonça et al. is simple. But we did not conclude about the performance. We think that this is well conveyed in the manuscript, and, for this reason, no further changes have been made.*

**RC. Minor**

**RC. - Thanks for adding Fig. 7 to test the relationship of muons on athmosperic data. To me, the large spread in the correlation to air pressure is suspicious, maybe this indicates the influence of factors to the signal other than cosmogenic muons? It may be worth a note.**

*AR: Thank you. We have added the following statement to acknowledge the possible influence of other factors.*

> The behaviour is attributed to the relative short time series and the small temperature range (±5°). However, the representativeness of air temperature measured at 2 m height in comparison to the need of a whole air temperature profile is also questionable and it should be further investigated (de Mendonça et al., 2016). The residual spread in the relationship suggests that the influence of factors to the signal other than cosmogenic muons cannot, however, be excluded and it should be considered in further studies.

**RC- Eq 3: Do you mean 1+alpha instead of 1-alpha? Have you corrected for air humidity following Rosolem et al. 2013 properly?**

*AR: Thank you. The equation in the manuscript was wrong and we corrected it in the new version of the manuscript. We double checked the analysis and we confirm that was implemented properly.*

**RC: - It is not clear from the methods description nor Figure 3 where the Medusa system was located.**

*AR: Thank you. We added this information in the text and in the caption of Figure 3.*

> For the assessment of the gamma signal measured by FINAPP3, a stationary CsI gamma-ray spectrometer (gSMS, Medusa Radiometrics, https://medusa-online.com/en/) has also been installed at Ceregnao site in 2021, few meters from the CRNS location.

> Figure 3. Experimental sites with FINAPP3 sensor (white points) and locations where gravimetric soil samples (red points) have been collected for comparison (pictures from Google Earth). At Ceregnano site, a gamma ray spectrometer was also installed few meters from the CRNS sensor.

*RC: - It is not clear from the methods description in what year the measurements have been taken. The whole presentation appeared a bit complicated to me, as many different types of data have been recorded during many different periods, even at the same sites, but also among different sites. I'd appreciate an overview either in text, plot, or table, to better streamline the readers understanding.*

*AR: Thank you for the comment. All the measurements have been collected in 2021. We have added this information in the new version of the manuscript where appropriated and in the caption. We briefly report below where this information is indicated, and we believe now the Readers should not be confused.*

*In L165 of the new version of the manuscript with track changes:*

> The recorded time series cover the period of seven months starting from May 2021 when, in both sites, a FINAPP3 detector was installed.

*In L183 of the new version of the manuscript with track changes:*

> A second assessment of the FINAPP3 sensor was carried out by a series of independent gravimetric soil sampling campaigns. The experiments were conducted in 2021 at four experimental sites located in the Po river plain, northern Italy (Figure 3).

*In L255 of the new version of the manuscript with track changes:*

> For the assessment of the gamma signal measured by FINAPP3, a stationary CsI gamma-ray spectrometer (gSMS, Medusa Radiometrics, https://medusa-online.com/en/) has also been installed at Ceregnao site in 2021, few meters from the CRNS location.

*RC: - The authors added a zoom on on the neutron data comparison and I am ackowledging it. While the point cloud in the correlation plot is not insightful (better use smaller points of non-filled points or transparency), the dynamics of the neutron data looks convincing. However, the CRS2000 and Lab-C data seem to be more stable (right panels c and f). Is this due to the higher count rate, or could it be that Finapp sees other influencing factors, too? Maybe the uncertainty bands could be indicated or mentioned in the caption.*

*AR: Thank you for the suggestions. The correlation plot has been improved to better visualize the results. The uncertainty bounds would cover a bit the visualization of the lines. For this reason, we did not add uncertainty bands to better appreciate the (un)stability in the signal as detected by the Reviewer. We believe that this behavior is due to the lower counting rate. For this reason, we have stressed in the manuscript that more sensitive detectors should be foreseen when fast hydrological changes should be detected (see L270). No further changes have been made to the text.*

*RC: - For a more conserverative formulation, I'd suggest to change in the conclusions "Muons were found to be a possible alternative for incoming correction for CRNS application" to "Muons were found to be a potential candidate to support the correction for incoming cosmic rays."*

*AR: Thank you. The text has been changed as suggested.*

> Muons were found to be a potential candidate to support the correction <s>a possible alternative</s> for incoming <s>correction for CRNS application</s>cosmic rays

**RC: - The sentence in the conclusion is misleading: "On the other hand, the use of gamma-ray spectrometry was identified as an alternative method for non-invasive soil moisture estimation.". I'd assume that you are referring to the cited paper who found this. But it was not found in your study as no spectrometer was used. Make more clear that this has been found by authors which is what motivated you to test a total gamma counter (which led to different findings).**

*AR: Yes, the Reviewer is right. The statement refers to previous studies where gamma ray spectrometers have been used. The text has been changed and the citations better placed to avoid misunderstanding as follows.*

> In previous studies, <s>Muons</s> muons were found to be a potential candidate to support the correction <s>a possible alternative</s> for incoming <s>correction for CRNS application</s>cosmic rays (Stevanato et al., 2022). On the other hand, the use of gamma-ray spectrometry was identified as an alternative method for non-invasive soil moisture estimation (Baldoncini et al., 2018) and irrigation discrimination (Serafini et al., 2021).

**RC: ## Cosmetics**

**RC: - It is very hard to identify dates within the plates for international readers. X tick marks like "03/07" are not very helpful and very confusing. Please use informative tick marks like "Jul 03". Also, most plots do not show a year, not even in the caption. It could be added to the first tick mark or to the caption.**

*AR: Thank you for the suggestion. We changed the X tick. Since all the data have been collected during the year 2021, we added this information to the caption. See the new version of the manuscript.*

**RC: - Most plots are very hard to read due to small panel size and greyish look. Avoid grey background, use clean, thin, black-and-white axis objects to increase the contrast, avoid colors whenever possible, choose meaningful axis ticks (see above), use column headings to highlight which panels correspond to which sites (e.g., Fig 8). Generally, consider using full-width plots for time series data instead of narrow panels that barely span 20% of the page width.**

*AR: We put some additional efforts to improve the quality of the plots and we hope now they have reached the right quality for publication.*

**References**

McJannet, D.L., Desilets, D., 2023. Incoming Neutron Flux Corrections for Cosmic-ray Soil and Snow Sensors Using the Global Neutron Monitor Network. Water Resources Research n/a, e2022WR033889. https://doi.org/10.1029/2022WR033889